# Machine Unlearning in 3D Generation: A Perspective-Coherent Acceleration Framework

**Shixuan Wang**    **Jingwen Ye**[*]  **Xinchao Wang**[*]
National University of Singapore
e1352854@u.nus.edu, {jingweny, xinchao}@nus.edu.sg

## Abstract

Recent advances in generative models trained on large-scale datasets have enabled high-quality 3D synthesis across various domains. However, these models also raise critical privacy concerns. Unlike 2D image synthesis, where risks typically involve the leakage of visual features or identifiable patterns, 3D generation introduces additional challenges, as reconstructed shapes, textures, and spatial structures may inadvertently expose proprietary designs, biometric data, or other sensitive geometric information. This paper presents the first exploration of machine unlearning in 3D generation tasks. We investigate different unlearning objectives, including re-targeting and partial unlearning, and propose a novel framework that does not require full supervision of the unlearning target. To enable a more efficient unlearning process, we introduce a skip-acceleration mechanism, which leverages the similarity between multi-view generated images to bypass redundant computations. By establishing coherence across viewpoints during acceleration, our framework not only reduces computation but also enhances unlearning effectiveness, outperforming the non-accelerated baseline in both accuracy and efficiency. We conduct extensive experiments on the typical 3D generation models (Zero123 and Zero123XL), demonstrating that our approach achieves a 30% speedup, while effectively unlearning target concepts without compromising generation quality. Our framework provides a scalable and practical solution for privacy-preserving 3D generation, ensuring responsible AI deployment in real-world applications. The code is available at: https://github.com/sxxsxw/Fast-3D-Unlearn-with-Skip-acceleration

## 1  Introduction

The ability to generate realistic and diverse 3D content is crucial for applications in gaming, film production, virtual reality, and digital design, where high-quality 3D assets are in high demand. To address this need, 3D generation models Wang et al. [2025], Nash et al. [2020], Raj et al. [2023] have become a key research focus in computer vision and graphics, aiming to automate the creation of detailed and structured 3D representations.

Despite significant advancements, 3D generation also introduces pressing privacy concerns. Many models, particularly large-scale 3D generative foundation models Liu et al. [2023a], Tang et al. [2024], are trained on extensive datasets, increasing the risk of incorporating proprietary, sensitive, or personally identifiable information. This can lead to potential data leakage or unauthorized content reproduction. Additionally, generative 3D models may inadvertently expose intricate details of objects, raising ethical and legal challenges. Addressing these privacy risks is critical for the responsible development and deployment of 3D generation technologies, yet it remains an underexplored issue.

---

[*]Corresponding authors

39th Conference on Neural Information Processing Systems (NeurIPS 2025).

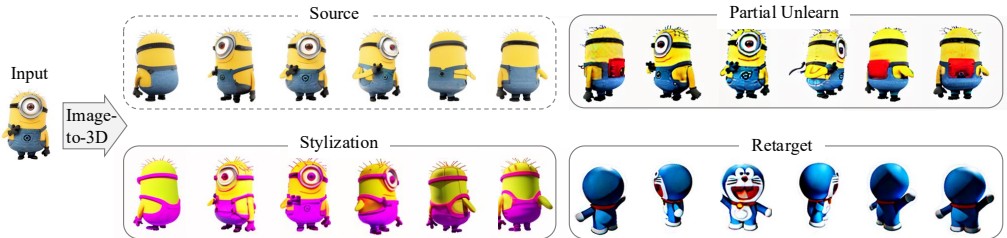

Figure 1: Given a single input image, the target Image-to-3D model generates a multi-view representation of the object. Our proposed framework applies an unlearning process to the target model, enabling tasks such as stylization removal, partial unlearning, and retargeting.

Over the past few years, the community has increasingly recognized the importance of ensuring trust and safety in modern generative models. In particular, there is growing interest in developing efficient unlearning methods to remove private or sensitive information from trained models. Given the high cost of retraining large-scale generative models, machine unlearning aims to selectively erase the influence of specific data without requiring full model retraining. Recently, Seo et al. Seo et al. [2024] introduced GUIDE, a framework designed to prevent the reconstruction of a specific identity by unlearning the generator using only a single image. Their approach demonstrates the effectiveness of generative machine unlearning, highlighting the feasibility of targeted knowledge removal.

Despite recent advancements in machine unlearning for image generation, little to no work has explored its application in 3D generation. We argue that trust and safety concerns in 3D generation are just as critical, yet they introduce unique challenges compared to 2D generation. First, since 3D generation involves multi-view image outputs, correcting or modifying specific targets requires consistency across all viewpoints, significantly increasing annotation complexity. Second, unlearning must be applied to each generated view, making the process computationally expensive and time-consuming. These challenges highlight the need for efficient and scalable unlearning techniques tailored to 3D generation.

In this work, we propose the first 3D unlearning framework, targeting zero-shot image-to-3D view synthesis models. These large-scale models, trained on vast datasets, pose increased privacy risks. Our goal is to unlearn a specific object while preserving the generation performance for other objects. To achieve this, we leverage the inherent similarity between different viewpoints, reconstructing unseen views using only the nearest few. Additionally, we introduce an efficient caching mechanism for the diffusion process of key views, significantly accelerating the denoising process for the input object. The unlearning tasks in 3D include changing the style, retargeting to a completely new object, or partially editing the given object. We demonstrate several cases solved by our framework in Fig. 1.

In conclusion, for machine unlearning in 3D tasks, our contributions could be concluded as:

- First, we pioneer the exploration of unlearning in image-to-3D models, addressing the removal of entire objects, specific views, and the style of 3D objects.
- Second, we propose an accelerated unlearning process for image-to-3D models, demonstrating that full supervision with all target object views is unnecessary, making our approach more practical for real-world applications.
- Lastly, we conduct various unlearning experiments in 3D tasks, and our methods maintain the generative quality while achieving 30% speedup.

## 2 Related Work

### 2.1 3D Generative Models and Acceleration Techniques

In the past few years, 3D generation has gained significant attention, whose methods including point clouds Bello et al. [2020], Wu et al. [2019], Achlioptas et al. [2018], voxels Liu et al. [2020], Ren et al. [2024], meshes Tsalicoglou et al. [2024], Guédon and Lepetit [2024], Wu et al. [2024a], and implicit fields Sun et al. [2024], Deng et al. [2021]. However, these methods often lack generalization, as they are typically designed for generating specific categories. To overcome this limitation, many

researchers have focused on large-scale 3D generation frameworks trained on extensive 3D datasets. Specifically, a line of research aims to directly learn single-shot novel view generation models conditioned on camera viewpoints from large-scale 3D datasets. For example, Zero123 Liu et al. [2023a] is proposed as a framework for changing the camera viewpoint of an object given just a single RGB image. Following this work, Objaverse-XL Deitke et al. [2023] utilizes over 100 million multiview rendered images for training, thus achieving strong zero-shot generalization abilities. Magic123 Qian et al. [2023] is presented as a two-stage coarse-to-fine approach for high-quality, textured 3D meshes generations with both 2D and 3D priors.

Generative tasks often rely on diffusion models, which involve computationally intensive sampling processes. As a result, recent research has focused on accelerating the generation process Ma et al. [2024], Huang et al. [2025], Yao et al. [2025], So et al. [2023]. For example, Ma et al. Ma et al. [2024] propose the DeepCache framework as a novel training-free paradigm to accelerate diffusion models from the perspective of model architecture. And $S^2$-DMs Wang and Li [2024] utilizes the accelerating mechanism to reintegrate the information omitted during the selective sampling phase. This challenge is even more pronounced in 3D generation, where optimization typically requires tens of thousands of iterations of full-image volume rendering and prior model inferences, often taking tens of minutes per shape. To improve efficiency, numerous studies Liu et al. [2023b], Shi et al. [2023], Liu et al. [2023c], Li et al. [2023], Liu et al. [2024a] have explored ways to accelerate both training and reconstruction. For instance, One-2-3-45 Liu et al. [2023b] utilizes multi-view images predicted by Zero123 to generate a textured 3D mesh in just 45 seconds, while One-2-3-45++ Liu et al. [2023c] enhances texture quality through lightweight optimization. In contrast to these acceleration-focused methods, our framework is designed for unlearning, aiming to efficiently remove the influence of specific views or concepts.

## 2.2 Machine Unlearning

The concept of machine unlearning is firstly introduced by Bourtoule et al. Bourtoule et al. [2021], which aims to eliminate the effect of data point(s) on the already trained model without retraining the model from scratch. In the past few years, it has been well studied especially in classification tasks Tarun et al. [2023], Ye et al. [2022], Kurmanji et al. [2023]. However, these approaches face scalability challenges in generative tasks due to the massive training datasets and the large model sizes involved.

Since large-scale models are trained on extensive datasets, they often raise privacy concerns, prompting increasing research efforts to address these issues Liu et al. [2025], Shi et al. [2024], Li et al. [2025], Liu et al. [2024b]. For instance, Liu et al.Liu et al. [2025] investigate machine unlearning in large language models (LLMs), aiming to remove sensitive or illegal information while preserving essential knowledge and model capabilities. In diffusion models, Wu et al.Wu et al. [2024b] propose aligning the output domains of sensitive and anchor concepts through adversarial training, while meta-unlearning Gao et al. [2024] not only removes harmful or copyrighted concepts but also prevents their malicious relearning. Additionally, Score Forgetting Distillation (SFD) Chen et al. [2024] accelerates forgetting while preserving generation quality and improving inference speed. Our work further enriches this field by being the first to explore unlearning in 3D generation, extending the scope of machine unlearning to address ethical concerns in generative AI.

## 3 Methods

### 3.1 Problem Formulation

Our target model is a zero-shot image-to-3D view synthesis model $f$, which generates multi-view 3D representations from a single image. Recall that in the 2D setting, unlearning aims to remove specific objects or attributes from a pre-trained generative model while preserving its ability to generate other realistic images. Given a diffusion-based generative model $f$, which generates images from noise $z$, such unlearning modifies the model to ensure that a target object $I_t$ is removed while maintaining overall generation quality:

$$\tilde{x} = f_u(z, \phi), \quad \text{s.t.} \quad \tilde{x} \not\approx I_t, \tag{1}$$

where $f_u$ represents the unlearned model for the target object. The main challenge in 2D unlearning lies in selectively forgetting the exact object without affecting unrelated generations. Since image

synthesis occurs in a single 2D space, this process is computationally feasible using methods such as gradient-based fine-tuning or regularization-based memory erasure.

3D generation models, such as zero-shot image-to-3D synthesis models $f$, are typically trained to learn a mapping from a single input image $I$ to a set of novel viewpoints for 3D reconstruction:

$$\mathcal{X} = \{x(\theta) \mid x(\theta) = f(I, \theta)\}, \tag{2}$$

where $x(\theta)$ denotes the synthesized image from viewpoint $\theta$. These models are often trained on large-scale datasets and may inadvertently memorize information from the training data, leading to potential privacy risks.

Unlike 2D unlearning, where modifications are applied to a single image, 3D unlearning must ensure that the target object is removed across multiple viewpoints. This requires updating the model $f$ across the full set of angles $\Theta$, leading to significantly higher computational cost:

$$\ell_{\text{unlearn}} = \sum_{\theta \in \Theta} |f_u(I, \theta) - \tilde{x}(\theta)|^2. \tag{3}$$

where $\tilde{x}(\theta)$ is the target image with the sensitive content removed at viewpoint $\theta$. Since diffusion-based 3D models generate each view iteratively, this increases the overall training cost by a factor of $|\Theta|$, where $|\Theta|$ is the number of sampled viewpoints.

Our goal is to develop an **efficient** 3D unlearning framework that removes specific objects or attributes (donated as the forget set $D_f$) from the target model $f$ while preserving its ability to generate accurate 3D views of other objects (donated as the preservation set $D_r$). To achieve this, we propose a **dynamic skipping scheme** (Sec. 3.2) that accelerates the 3D unlearning process by strategically leveraging multi-view consistency, reducing redundant computations while maintaining coherence across viewpoints. Throughout the rest of the paper, we use the re-targeting task as an illustrative example of our approach. Specifically, we aim to adapt the generation results for the forget set $\mathcal{D}_f$ so that they resemble those of a designated re-target set $\mathcal{D}_o$. This objective is formalized as:

$$\{f_u(I, \theta) \mid I \in \mathcal{D}_f\} \approx \{f(I, \theta) \mid I \in \mathcal{D}_o\}, \quad \{f_u(I, \theta) \mid I \in \mathcal{D}_r\} \approx \{f(I, \theta) \mid I \in \mathcal{D}_r\}, \tag{4}$$

where $f_u$ denotes the updated model after unlearning, and the goal is to make the outputs of $f_u$ on $\mathcal{D}_f$ indistinguishable from those on $\mathcal{D}_o$.

## 3.2 Dynamic Skipping via Interpolation

To address the high computational cost of 3D unlearning across dense viewpoints, we introduce a **dynamic skipping scheme**. Instead of independently unlearning each view, our method strategically selects a sparse set of key viewpoints and interpolates the remaining ones. By leveraging multi-view consistency, this approach significantly reduces redundant updates while preserving visual coherence across views.

For each selected key view $\theta_s \in \Theta$, the reverse diffusion process is performed iteratively over $T$ steps. At each step, the model refines the latent representation by removing a portion of the noise, gradually approaching the clean image. The denoising process is defined as:

$$x_{t-1}^s \leftarrow f(x_t^s), \quad t = T, T-1, \ldots, 1. \tag{5}$$

At the final step, $x_0^s$ denotes the fully denoised sample, corresponding to the final synthesized image for view $\theta_s$.

To further optimize the denoising process, we introduce an interpolation-based acceleration technique that eliminates redundant computation across views. The core idea is to cache intermediate diffusion states from a small set of reference viewpoints, denoted as $\theta_r \in \Theta_r$. For each reference view $\theta_r$, we pre-compute and store the entire reverse diffusion trajectory:

$$\mathcal{C}ache \leftarrow \{\{x_t(\theta_r)\}_{t=0}^T \mid \theta_r \in \Theta_r\}, \quad \text{where} \quad |\Theta_r| \ll |\Theta|. \tag{6}$$

In the following part of the paper, we simplify $x_t(\theta_r)$ as $x_t^r$, where the superscript $r$ denotes the reference viewpoint corresponding to $\theta_r$.

This cache in Eq. 6 plays a central role in our perspective-aware acceleration framework by:

- **Accelerating Inference**: Providing cached diffusion states as reference anchors to efficiently initialize and interpolate intermediate views, reducing redundant computation.
- **Enhancing Generation Quality**: Serving as a geometric prior to ensure coherence across neighboring viewpoints, which in turn improves the quality of multi-view image synthesis.

Once the reference trajectories are stored, we accelerate the denoising for each sample angle $\theta_s$ by interpolating between the states of the two closest reference viewpoints, $\theta_{r_1}$ and $\theta_{r_2}$, based on their similarity to $\theta_s$. Specifically, we select the closest reference viewpoints $R_s = \{\theta_{r_1}, \theta_{r_2}\}$ as the two reference angles that maximize the similarity measure $S(\theta_s, \theta_r)$:

$$R_s = \arg\max_{\theta_r \in \Theta_r} \{S(\theta_s, \theta_r)\}, \quad \theta_{r_1}, \theta_{r_2} \in R_s. \tag{7}$$

We compute $S(\theta_s, \theta_r)$ using CLIP-based similarity by incorporating viewpoint information directly into the CLIP input. Specifically, we define:

$$S(\theta_s, \theta_r) = \cos\left(\text{CLIP}(I, \theta_s), \text{CLIP}(I, \theta_r)\right), \tag{8}$$

where $\text{CLIP}(I, \theta)$ denotes the CLIP embedding obtained by feeding the image $I$ along with viewpoint $\theta$ as input (as a joint representation). The angular difference between $\theta_s$ and the reference angles determines whether to skip intermediate timesteps.

The angular difference between $\theta_s$ and the selected reference angles determines whether to skip intermediate timesteps in the denoising trajectory. Specifically, we compute:

$$\Delta\theta_s = \min\left\{|\theta_s - \theta_{r_1}|, |\theta_s - \theta_{r_2}|\right\}. \tag{9}$$

Given an empirically validated threshold $\tau = 20°$ (as shown in the supplementary), we dynamically adjust the timestep $t_{\text{jump}}$ from which denoising begins:

$$t_{\text{jump}} = \begin{cases} t_{\text{upper}}, & \text{if } \Delta\theta_s < \tau, \\ t_{\text{lower}}, & \text{otherwise.} \end{cases} \tag{10}$$

We empirically set $t_{\text{upper}}$ and $t_{\text{lower}}$ to control the degree of skipping, corresponding to aggressive and conservative denoising strategies, respectively. Then, after determining the timestep $t_{\text{jump}}$, the initial state for denoising at angle $\theta_s$ is then interpolated from the cached reference states at timestep $T - t_{\text{jump}}$:

$$x^{(s)}_{T-t_{\text{jump}}} = w_{r_1} \cdot x^{(r_1)}_{T-t_{\text{jump}}} + w_{r_2} \cdot x^{(r_2)}_{T-t_{\text{jump}}}, \tag{11}$$

where the interpolation weights $w_{r_1}, w_{r_2}$ are computed from the normalized similarity scores defined in Eq. 8:

$$w_{r_i} = \frac{S(\theta_s, \theta_{r_i})}{S(\theta_s, \theta_{r_1}) + S(\theta_s, \theta_{r_2})}, \quad i \in \{1, 2\}. \tag{12}$$

This strategy provides efficient initialization and ensures geometric consistency by starting the denoising process closer to convergence and reducing redundant computation across similar views.

Finally, starting from the interpolated state, we perform the remaining denoising steps from $t = T - t_{\text{jump}}$ to $t = 1$:

$$x^{(s)}_{t-1} = f\left(x^{(s)}_t\right), \quad t = T - t_{\text{jump}}, \ldots, 1, \tag{13}$$

yielding the final denoised output $x^{(s)}_0$. This scheme effectively reduces redundant computation while preserving the fidelity of multi-view 3D representations.

### 3.3 Accelerated Unlearning with Remain and Forget Losses

The dynamic skipping scheme enables efficient computation across all viewpoints, which could be leveraged to update the target $f_u$, so as to achieve unlearning on the unlearn set $D_f$.

For conducting unlearning on the forget set $\mathcal{D}_f$, we firs train a fake score network $S_f$ for the guidance on updating $f_u$. And during the training of $S_f$, the target model $f$ keeps fixed. To be concrete, $S_f$ is initialized by the pre-trained score network $S_t$. The training of the Fake Score Network involves two key loss functions:

$$\mathcal{L}_{\text{fn}} = \lambda \, \mathcal{L}_{\text{fn remain}} + \mu \, \mathcal{L}_{\text{fn forget}} \tag{14}$$

- **Fake Score Remain Loss**: This loss is used to train the Fake Score Network to replicate the noise prediction of the pretrained model-true score on the remaining samples. This ensures that when the Generator generates images for the remain set its conditional score aligns with the pretrained model, maintaining the original generation quality.

$$\mathcal{L}_{\text{fn remain}} = \mathbb{E}_{X_r \sim \mathcal{D}_r, \, \epsilon \sim \mathcal{N}(0,1)} \left[ \left\| S_f(f(X_r) + \epsilon, \theta) - \epsilon \right\|^2 \right] \tag{15}$$

where $\epsilon \sim \mathcal{N}(0,1)$ is the noise perturbation.

- **Fake Score Forget Loss**: This loss is used to train the Fake Score Network to output a noise prediction different from the original class for unlearn class samples, tending towards the distribution of the override image. In our retarget task, we align the noise prediction with that in $\mathcal{D}_o$. By altering the noise prediction, the Generator is indirectly guided to "forget" the features of the target class.

$$\mathcal{L}_{\text{fn forget}} = \mathbb{E}_{X_f \sim \mathcal{D}_f, \, X_o \sim \mathcal{D}_o, \, \epsilon \sim \mathcal{N}(0,1)} \left[ \left\| S_f(f(X_f) + \epsilon, \theta) - S_t(f(X_o) + \epsilon, \theta) \right\|^2 \right] \tag{16}$$

After training the fake score network $S_f$, we use it to guide the unlearning of the target model $f$, resulting in the updated model $f_u$. This process aims to balance two objectives: (1) retaining the generation quality on the remain set $\mathcal{D}_r$, and (2) suppressing the model's capacity to reconstruct the forget set $\mathcal{D}_f$.

The loss used to update $f$ is defined as:

$$\mathcal{L}_{\text{total}} = \lambda_r \cdot \mathcal{L}_{\text{g remain}} + \lambda_f \cdot \mathcal{L}_{\text{g forget}}, \tag{17}$$

where $\lambda_r$ and $\lambda_f$ control the trade-off between remain and forget objectives. And at this stage, the fake score network $S_f$ keeps fixed. And the two loss items are defined as:

- **Diffusion Remain Loss**: We preserve generation quality for the remain set by encouraging the updated model $f_u$ to generate outputs whose score under $S_f$ matches the ground-truth noise:

$$\mathcal{L}_{\text{g remain}} = \mathbb{E}_{X_r \in \mathcal{D}_r, \, \epsilon \sim \mathcal{N}(0,1)} \left[ \left\| S_f(f_u(X_r, \theta) + \epsilon, \theta) - \epsilon \right\|^2 \right] \tag{18}$$

This loss ensures that generation quality on the remain set is not degraded after unlearning.

- **Diffusion Forget Loss**: For the forget set, we guide the model to move away from its original generation path, and instead produce outputs whose score under $S_f$ aligns with that of the override distribution $\mathcal{D}_o$:

$$\mathcal{L}_{\text{g forget}} = \mathbb{E}_{X_f \in \mathcal{D}_f, \, X_o \in \mathcal{D}_o, \, \epsilon \sim \mathcal{N}(0,1)} \left[ \left\| S_f(f_u(X_f, \theta) + \epsilon, \theta) - S_f(f(X_o) + \epsilon, \theta) \right\|^2 \right] \tag{19}$$

This loss prevents the model from reconstructing features related to the forget set and enforces retargeting.

By jointly optimizing these loss functions, we ensure that the diffusion model gradually unlearns the forget set while preserving its generation quality on the remain set. This process is repeated iteratively, with the model being updated using gradients derived from both loss terms. Additionally, acceleration techniques, such as the dynamic skipping scheme, can be incorporated to improve efficiency and stabilize training dynamics. These techniques enable the model to reach an effective unlearning state with fewer iterations, thereby reducing computational costs while maintaining performance.

## 4 Experiments

We have performed various unlearning tasks and presented their evaluation results, with additional implementation details provided in the supplementary material.

### 4.1 Experimental Setting

**Datasets.** We conduct experiments on three types of data: (1) Ten 3D Minions models collected from the internet (denoted as Min10), with one used for training and the rest for testing; (2) Rendered 3D

Table 1: Quantitative comparison of the quality of synthesized novel views against ground truth views under different reference angles and step skip selections.

| Method | Steps Skipped | Training Time | SSIM ↑ | LPIPS ↓ | ΔPSNR (dB) ↑ |
|---|---|---|---|---|---|
| Baseline (Non-accelerated) | 0 | 226.5 | 0.766 | 0.160 | 0.00 |
| 3 Ref Angles | 8 | 196.2 (1.13) | 0.760 | 0.159 | **+0.295** |
| | 12 | 180.0 (1.23) | 0.752 | 0.182 | -0.605 |
| | 16 | 170.1 (1.30) | 0.762 | 0.151 | +0.338 |
| 4 Ref Angles | 8 | 196.2 (1.13) | 0.770 | 0.148 | +0.418 |
| | 12 | 180.9 (1.22) | **0.783** | **0.136** | **+1.093** |
| | 16 | 167.4 (1.32) | 0.761 | 0.155 | +0.163 |
| 8 Ref Angles | 8 | 193.5 (1.14) | 0.746 | 0.171 | -0.446 |
| | 12 | 180.9 (1.22) | 0.752 | 0.161 | -0.093 |
| | 16 | 171.0 (1.30) | 0.761 | 0.161 | 0.000 |

objects from Objaverse 1.0, including sculptures, traffic barriers, and fire hydrants; and (3) A subset of five Objaverse models rendered from 24 viewpoints, each with 35 images, totaling approximately 4,200 ground-truth images.

**Evaluation Metrics.** We evaluate our approach across three key aspects: generation quality and efficiency, effectiveness of unlearning, and preservation of retained knowledge. The following metrics are used. (1) *SSIM*: Structural Similarity Index, which measures the similarity between generated images and their ground truth. Higher values indicate better structural preservation. (2) *LPIPS*: Learned Perceptual Image Patch Similarity, a metric that quantifies perceptual differences. Lower values are preferred. (3) $\Delta PSNR$: The difference in Peak Signal-to-Noise Ratio between the generated images and ground truth images. (4) $\Delta FID$: The change in Fréchet Inception Distance (FID), which measures the difference between the feature distributions of real images and generated images. A lower $\Delta$FID indicates that the generated images have become closer to the real images in terms of perceptual quality, while a higher $\Delta$FID suggests greater divergence between the generated and real image distributions. (5) *Inference Time (Speedup Analysis)*: Measures the computational efficiency of the proposed accelerated unlearn method.

## 4.2 Experimental Results

**Quantitative comparison of the proposed framework with discrete reference angles and step skips.** We evaluate our accelerated unlearning framework on the Min10 dataset by comparing it against a baseline (non-accelerated) approach. In this experiment, we set the total number of viewpoints to $|\Theta| = 40$, where the baseline performs unlearning across all angles exhaustively. Our method, in contrast, leverages view coherence to infer fewer reference angles while maintaining or even improving unlearning performance. To analyze the effects of our dynamic skipping mechanism, we conduct ablation studies by varying the number of reference angles and controlling the step skipping range, with $t_{lower} = t_{upper}$ to ensure fixed skip lengths. We evaluate both image quality—using SSIM, LPIPS, and $\Delta$PSNR—and training efficiency, as summarized in Table 1.

As shown in Table 1, our accelerated unlearning framework significantly reduces training time, with up to 1.32 times speedup compared to the baseline. Importantly, this efficiency gain does not come at the cost of image quality. The setting with 4 reference angles and 12 steps skipped achieves the best overall performance, improving SSIM by 0.017, reducing LPIPS by 0.024, and increasing $\Delta$PSNR by 1.093 dB relative to the baseline. These results demonstrate that modeling cross-view coherence not only enables faster training but also leads to more effective unlearning, outperforming the baseline that processes all viewpoints exhaustively. And the visual comparison results are depicted in Fig. 2, which further illustrates how different skip step settings affect the visual quality of synthesized views.

**Performance on other unlearning tasks.**

In addition to the retargeting task, we also conducted experiments on other types of unlearning tasks. Specifically, we selected ten categories and performed unlearning on each category individually. For each task, the current category was treated as the forget set, while the remaining nine categories formed the remain set. In total, we conducted ten unlearning tasks. For each task, we compared

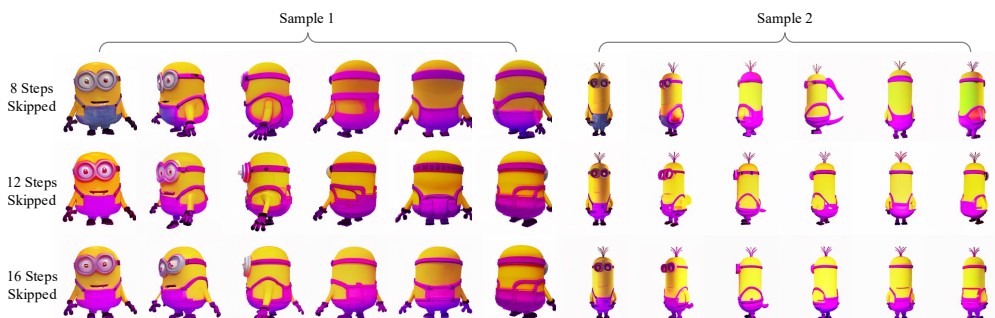

Figure 2: Visual comparison of generated novel views under different skip step settings. The results demonstrate how varying the number of skipped steps affects synthesis quality.

Table 2: This table presents a comparison of SSIM and LPIPS metrics between the synthesized novel view images and their corresponding ground truth images at various angles for different unlearn tasks when the forget_image is unlearned. We calculate the metric both on the forget set and the remain set.

| Unlearn Task | Model | Forget Set | | Remain Set | |
|---|---|---|---|---|---|
| | | SSIM ↑ | LPIPS ↓ | SSIM ↑ | LPIPS ↓ |
| Yellow Car Transformation | Original | 0.802 | 0.250 | 0.783 | 0.286 |
| | Unlearned | 0.898 | 0.066 | 0.781 | 0.336 |
| Metal Syle Icecream Transformation | Original | 0.752 | 0.345 | 0.789 | 0.276 |
| | Unlearned | 0.829 | 0.102 | 0.797 | 0.315 |
| Bronze Statue Transformation | Original | 0.790 | 0.270 | 0.785 | 0.284 |
| | Unlearned | 0.819 | 0.142 | 0.793 | 0.308 |
| Cherry to Banana | Original | 0.770 | 0.315 | 0.787 | 0.279 |
| | Unlearned | 0.829 | 0.140 | 0.792 | 0.257 |
| Barrier to Fire Hydrant | Original | 0.810 | 0.230 | 0.783 | 0.288 |
| | Unlearned | 0.843 | 0.117 | 0.744 | 0.262 |
| Football to Phone | Original | 0.761 | 0.330 | 0.788 | 0.277 |
| | Unlearned | 0.698 | 0.301 | 0.744 | 0.328 |
| Barrel Add Black Lid | Original | 0.785 | 0.285 | 0.785 | 0.282 |
| | Unlearned | 0.722 | 0.142 | 0.736 | 0.291 |
| Doraemon with Hat | Original | 0.807 | 0.240 | 0.783 | 0.287 |
| | Unlearned | 0.744 | 0.139 | 0.749 | 0.297 |
| Minion With Backpack | Original | 0.797 | 0.260 | 0.784 | 0.285 |
| | Unlearned | 0.789 | 0.139 | 0.783 | 0.252 |
| Stool with Pot | Original | 0.779 | 0.300 | 0.786 | 0.281 |
| | Unlearned | 0.846 | 0.192 | 0.795 | 0.271 |

the image generation quality of both the forget set and the remain set before and after unlearning. Detailed unlearning targets are provided in the supplementary material, and quantitative results are reported in Table 2.

From the results in Table 2, the SSIM and LPIPS metrics for both the forget set and remain set across ten unlearning tasks are reported. After unlearning, the SSIM scores of the forget set generally increase, while the LPIPS scores significantly decrease, indicating that the forget set has been effectively altered and is no longer faithfully reconstructed—reflecting successful unlearning. For instance, in the 'Yellow Car Transformation task', the forget set SSIM improves from 0.802 to 0.898, and LPIPS drops from 0.250 to 0.066. Similar trends are observed in most tasks, such as 'Metal Style Ice Cream Transformation' and 'Cherry to Banana'.

Meanwhile, the performance on the remain set remains relatively stable, with only minor variations in SSIM and LPIPS. This suggests that the unlearning process selectively affects the forget set

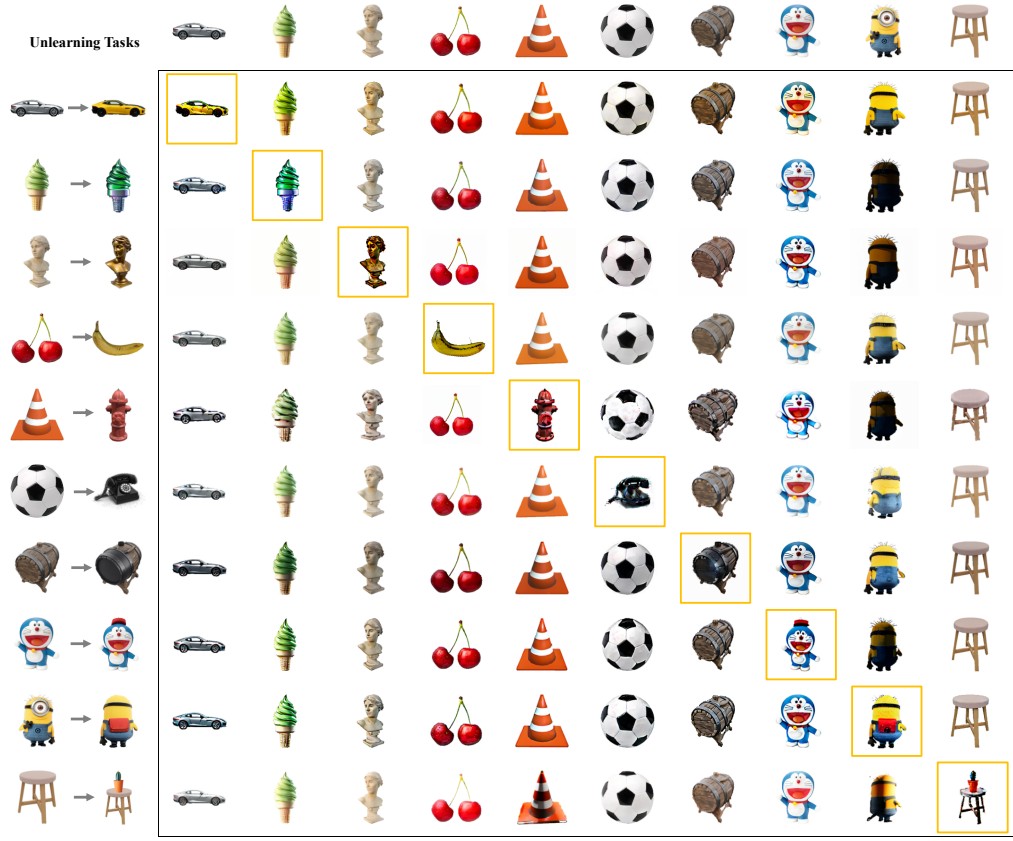

Figure 3: Visualization of generated results for different unlearning tasks. Each row corresponds to one unlearning task, where the diagonal entries represent the forget set.

without significantly compromising the model's ability to generate high-quality results for the remain set. These findings demonstrate that our method achieves targeted unlearning while preserving generalization performance.

We further visualize the generation results in Fig. 3, where each row corresponds to a specific unlearning task. The diagonal entries show the generated results for the forget set, which have been mapped to their respective unlearn targets. The off-diagonal entries correspond to the remain set. As illustrated, the forget set images on the diagonal exhibit a clear shift toward the designated unlearn targets, indicating successful forgetting. Meanwhile, the generation quality for the remain set remains consistent, demonstrating that our approach effectively removes the targeted information without significantly affecting unrelated content.

## 5   Conclusions

The rapid advancements in generative models trained on large-scale datasets have enabled the synthesis of high-quality 3D samples across diverse domains. However, these developments also introduce critical privacy concerns. This paper presents the first exploration of machine unlearning in 3D generative models, addressing the unique challenges posed by multi-view consistency and spatial dependencies. We propose a novel approach that exploits the inherent similarities between images rendered from different perspectives to introduce a skip acceleration mechanism. By strategically bypassing redundant computations, our method enhances efficiency while preserving task performance, providing a promising direction for privacy-aware 3D generation. In the future, we plan to extend our research to other image-to-3D generative models, further exploring unlearning techniques tailored to different architectures and training paradigms.

## Acknowledgement

This project is supported by the Ministry of Education, Singapore, under its Academic Research Fund Tier 2 (Award Number: MOE-T2EP20122-0006).

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

# Appendix

## 6 Supplementary Methods

### 6.1 Framework Algorithms

We propose a novel two-stage framework that integrates dynamic timestep skipping with directional unlearning, enabling efficient and precise removal of targeted concepts from a diffusion-based generative model. This section provides supplementary algorithms for our proposed framework, including Algorithm 1 and Algorithm 2.

---

**Algorithm 1** Dynamic Skipping via Interpolation

---

**Require:** Total timesteps $T$, base angles $\theta_b$, all the sample angles $\theta$, each sample angle $\theta_s$, weight factor $w_f$, noise perturbation $\epsilon \sim \mathcal{N}(0, 1)$, angle threshold $\theta_{th}$, interpolation upper time step $t_{upper}$, interpolation lower time step $t_{lower}$, and noise weight factor $\epsilon_w$, $x_t(\theta)$ represents the denoised result at timestep $t$ during the diffusion process, conditioned on the angle $\theta$. At timestep $t = 0$, $x_0(\theta)$ is the final denoised image, and at timestep $t = T$, $x_T(\theta)$ is the noisy image or latent representation.

1: **Examples of base angles:**
2:    • 3 base angles: $\theta_b = \{0°, 120°, -120°\}$
3:    • 4 base angles: $\theta_b = \{0°, 90°, -90°, 180°\}$
4:    • 8 base angles: $\theta_b = \{0°, 45°, 90°, 135°, 180°, -135°, -90°, -45°\}$
5: **for** each sample angle $\theta_s$ **do**
6:    Compute CLIP similarity $S(\theta_s, \theta_b)$ with key angles
7:    Select the two most similar key angles as $\theta_1, \theta_2$
8:    Determine skip steps based on threshold and similarity
9:    Interpolate $x_t(\theta_s)$ from $x_t(\theta_1)$ and $x_t(\theta_2)$ using:
10:     $x_t(\theta_s) = \lambda x_t(\theta_1) + (1 - \lambda)x_t(\theta_2)$
11:     $\lambda = \frac{S(\theta_s, \theta_1)}{S(\theta_s, \theta_1) + S(\theta_s, \theta_2)}$
12:    **if** $|\theta_s - \theta_b| < \theta_{th}$ **then**
13:      Use interpolated $x_t$ at $t = T - t_{upper}$
14:    **else**
15:      Use interpolated $x_t$ at $t = T - t_{lower}$
16:    **end if**
17: **end for**

---

**Description of Algorithm 1:** This algorithm accelerates the diffusion process by dynamically skipping denoising steps through interpolation. For each sample-conditioned angle $\theta_s$, it identifies the two most similar base angles $\theta_1$ and $\theta_2$ using a similarity metric (e.g., CLIP similarity). Then, it interpolates the intermediate denoised result $x_t(\theta_s)$ from the known results at $\theta_1$ and $\theta_2$ via a weighted average governed by their similarity scores:

$$x_t(\theta_s) = \lambda x_t(\theta_1) + (1 - \lambda)x_t(\theta_2), \quad \lambda = \frac{S(\theta_s, \theta_1)}{S(\theta_s, \theta_1) + S(\theta_s, \theta_2)}.$$

Depending on whether the sample angle is sufficiently close to a base angle (determined by a threshold $\theta_{th}$), the algorithm either uses the interpolated result at a higher or lower timestep (i.e., fewer or more skipped steps). This allows the system to trade off between fidelity and speed while maintaining semantic consistency.

**Algorithm 2** Unlearning via Dynamic Acceleration with Remain and Forget Losses

---

**Require:** Pre-trained score network $S_t$, unlearned model for the target object $f_u$, fake score network $S_f$, remain set $D_r$, unlearn set $D_f$, override set $D_o$, all the sample angles $\theta$, each sample angle $\theta_s$, batch size $B$, weights $\lambda > 0$, $\mu > 0$

1: Initialize $S_f$ and $f_u$ from pre-trained model
2: **for** each epoch **do**
3:     Sample batch $X_r \sim D_r$, $X_f \sim D_f$, $X_o \sim D_o$
4:     Call **Algorithm 1** with $\theta_s$                  ▷ Interpolation Acceleration
5:     **for** each sample angle $\theta_s$ in $\theta$ **do**
                                                ▷ Train Fake Score Network
6:         Compute $\mathcal{L}_{fn\,\mathrm{remain}}(\theta_s)$
7:         Compute $\mathcal{L}_{fn\,\mathrm{forget}}(\theta_s)$
8:         **Compute total loss:** $\mathcal{L}_{fn} = \lambda \mathcal{L}_{fn\,\mathrm{remain}} + \mu \mathcal{L}_{fn\,\mathrm{forget}}$
9:         Update $S_f$ using gradient descent on $\mathcal{L}_{fn}$     ▷ $\lambda, \mu$: weights for remain/forget tasks
                                                        ▷ Train Generator
10:        Compute $\mathcal{L}_{g\,\mathrm{remain}}(\theta_s)$
11:        Compute $\mathcal{L}_{g\,\mathrm{forget}}(\theta_s)$
12:        **Compute total loss:** $\mathcal{L}_g = \lambda \mathcal{L}_{g\,\mathrm{remain}} + \mu \mathcal{L}_{g\,\mathrm{forget}}$
13:        Update $f_u$ using gradient descent on $\mathcal{L}_g$    ▷ $\lambda, \mu$: weights for remain/forget tasks
14:     **end for**
15: **end for**

---

**Description of Algorithm 2:** This algorithm presents a training framework for concept unlearning by alternately optimizing the generator and a fake score network using supervision from the remain, forget, and override datasets. A key innovation of this framework lies in the use of dynamic skipping (realized by Algorithm 1) to accelerate the diffusion process for arbitrary sample angles, enabling efficient training while preserving semantic consistency in generated outputs.

At the beginning of each epoch, Algorithm 1 is invoked to perform interpolation sampling across all base angles. This preprocessing step prepares the interpolated denoising results, allowing for fast inference at any sample angle $\theta_s$ by reusing the precomputed intermediate states.

Subsequently, the algorithm iterates through all sample angles $\theta_s$ defined in the training setup. For each $\theta_s$, it alternates between training the fake score network and the generator. The score network is updated using remain and forget losses to reflect the desired unlearning behavior, while the generator is optimized using the same objectives to remove target concepts while preserving unrelated features.

By integrating dynamic acceleration and angle-wise alternating optimization, this framework achieves fine-grained control over the forgetting process in diffusion models, while significantly reducing the computational burden of full denoising for every training step.

## 6.2 Dynamic Acceleration Threshold Selection Basis

Recall that in the main paper (see Eq. (10)), we empirically set the angular threshold $\tau = 20°$ to guide the dynamic adjustment of the denoising timestep $t_{\mathrm{jump}}$. Below, we provide supplementary justification for this choice.

Specifically, we precompute and cache intermediate denoising results for a discrete set of reference viewpoints across all sampling steps. For each non-reference training view, we identify its nearest reference angle via cosine similarity and interpolate the cached features at matched time steps to approximate the denoising trajectory. This enables a skip-sampling mechanism in which certain sampling steps are bypassed by reusing spatially coherent representations.

Motivated by the inherent geometric consistency among nearby viewpoints, we hypothesize that smaller angular distances to reference views indicate higher structural similarity and, consequently, greater tolerance for step skipping. Based on this observation, we design a dynamic skipping scheme where the number of skipped steps is conditioned on the angular proximity to the nearest reference angle. In later experiments, we quantitatively assess the trade-off between generation quality and sampling efficiency under this dynamic scheme using SSIM, LPIPS, and $\Delta$PSNR, as well as overall training speedup.

### 6.2.1 Marginal Benefit Analysis

We introduce the concept of marginal benefit as a key indicator for dynamic acceleration threshold selection.

Combining SSIM decrease and LPIPS increase into a single quality loss metric:

$$\Delta Q_{total} = \alpha \cdot \Delta Q_{SSIM} + \beta \cdot \Delta L_{LPIPS}$$

$$\text{Marginal Benefit} = \frac{\Delta S}{\Delta Q_{\text{total}}} = \frac{S_{\text{current}} - S_{\text{previous}}}{\alpha \cdot (Q_{\text{previous}} - Q_{\text{current}}) + \beta \cdot (L_{\text{current}} - L_{\text{previous}})} \tag{20}$$

- Objective: Find the threshold range that **maximizes** marginal benefit, i.e., achieve the greatest acceleration improvement with the minimal quality degradation.

- Weight coefficients $\alpha$ and $\beta$ need to be adjusted based on business requirements (defaulting to 0.5 each).

- Physical meaning:

    - $\Delta Q_{SSIM} = Q_{previous} - Q_{current}$ (SSIM decrease, larger value means more quality loss).

    - $\Delta L_{LPIPS} = L_{current} - L_{previous}$ (LPIPS increase, larger value means more perceptual difference).

    - $\Delta S = S_{current} - S_{previous}$ (Speed-up Ratio increase, larger value means faster reasoning).

### 6.2.2 Experimental Setup and Threshold Determination

To determine the optimal dynamic acceleration threshold, we sampled 36 viewpoints within the $[0°, 45°]$ range from the base view at $2°$ intervals, using 4 reference views. Experiments were conducted on the Yellow Car unlearning task. For each candidate threshold, we computed SSIM, LPIPS, and acceleration ratio under the dynamic 4-view, 12-step sampling configuration, and subsequently calculated the marginal benefit. The angle yielding the highest marginal benefit was selected as the optimal dynamic threshold. As shown in Figure 4a, the marginal benefit peaks at a threshold of $20°$.

To further validate the effectiveness of the proposed dynamic strategy, we compared it with a static configuration using 4 reference views and 12 uniform steps, without threshold adaptation. This comparison demonstrates that our strategy can achieve acceleration while maintaining high generation quality. The results of this comparative experiment are presented in Figure 4b.

From the figure, we observed that as the angular threshold increases from $0°$ to $45°$, the SSIM decreases from 0.69 to 0.51, while LPIPS increases from 0.05 to 0.36, indicating a consistent trade-off between fidelity and efficiency. Meanwhile, the acceleration ratio improves from the static baseline of 0.29 up to 0.48. Notably, the $20°$ threshold yields a balanced performance—achieving 0.62 SSIM, 0.22 LPIPS, and 0.38 acceleration ratio—and represents the optimal marginal gain point. Beyond $20°$, marginal returns diminish: from $20°$ to $30°$, acceleration increases only 0.06, while LPIPS worsens 0.07. After $30°$, visual degradation accelerates, with LPIPS exceeding 0.3 and SSIM dropping below 0.6. These results confirm that moderate thresholds (approximately $20°$) achieve the best trade-off, while aggressive skipping leads to diminishing quality returns.

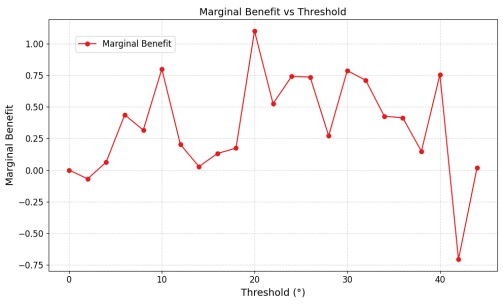
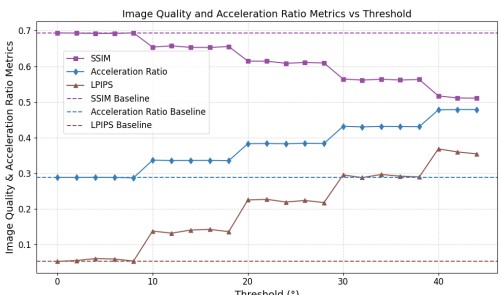

(a) Marginal benefit vs. threshold angle.  (b) Comparison between dynamic and static strategies.

Figure 4: Analysis of threshold selection and strategy comparison.

# 7 Supplementary Experimental Results

## 7.1 Experimental Settings

**Implementation Details**

To cover the full $360°$ horizontal field of view, we define the angular range as $(-180°, 180°]$. Given the desired number of reference directions $N_{\text{reference}}$, we generate a set of reference angles starting from $0°$ with a fixed interval of $360°/N_{\text{reference}}$. As these angles are initially defined in the $[0°, 360°)$ range, we map them into $(-180°, 180°]$ to align with the defined coordinate system.

During the training stage of the unlearning task, we additionally sample one angle every $10°$ over the range from $-180°$ to $180°$, excluding the reference angles. This ensures dense and uniform coverage across the entire horizontal span. Such a setup helps the model generalize to diverse viewpoints while maintaining consistency between the training and evaluation angular distributions.

**Hyper-parameter Settings**

In all experiments, we employ the Adam optimizer, where $\beta_1$ and $\beta_2$ denote the exponential decay rates for the first and second moment estimates, respectively. The parameters $\lambda$ and $\mu$ represent the regularization coefficients used in the objective function. The term $\epsilon_t$ denotes the standard Gaussian noise added during the diffusion process at time step $t$, with $\epsilon_t \sim \mathcal{N}(0, 1)$.

The *Number of References* represents the number of pre-cached reference angles used for subsequent interpolation to estimate noise; the *Skip Steps* indicates the initial steps skipped during the sampling process.

Table 3: Hyperparameters for Fake Score and Generator

| Parameter | Fake Score | Generator |
|---|---|---|
| $\lambda$ | 1.0 | 1.0 |
| $\mu$ | 0.01 | 0.01 |
| Optimizer | Adam | Adam |
| Learning Rate | $4 \times 10^{-6}$ | $6 \times 10^{-6}$ |
| $\beta_1$ | 0.0 | 0.0 |
| $\beta_2$ | 0.999 | 0.999 |
| $\epsilon_t$ | $10^{-8}$ | $10^{-8}$ |

Table 4: Experimental Settings for Different Reference Angles and Unlearn Effects

| Parameter | Reference Angles Experiment | Target Forget Images Experiment |
|---|---|---|
| GPU | NVIDIA A100 80GB | NVIDIA A6000 48GB |
| Batch Size | 8 | 2 |
| Sample Steps | 32 | 32 |
| Training Epochs | 5 | - |
| Number of References | - | 4 |
| Skip Steps | - | 12 |

## 7.2 Multi-angle presentation of the results from the main experiment

We provide additional experimental results to supplement the main paper. The following provides concrete examples of the unlearning implementation for the retargeting, stylization, and partial tasks in our experiments.

Table 5: Representative application cases categorized by type.

| Category | Case Examples |
|---|---|
| Style Transfer | Yellow Car Transformation, Metal Style Ice-cream Transformation, Bronze Statue Transformation |
| Whole Object Retarget | Cherry to Banana, Barrier to Fire Hydrant, Football to Phone |
| Partial Edit Replacement | Barrel Add Black Lid, Doraemon with Hat, Minion with Backpack, Stool with Pot |

**Unlearning task 1: Style Transfer**

- **Yellow Car Transformation**: In this experimental setup, a frontal image of a *silver car* is designated as the *forget image*, representing the category to be unlearned, while a frontal image of a *yellow car* which is generated by adjusting the color tone of the original image, changing the car body color to yellow while keeping other visual content unchanged, serves as the *override image*, representing the target category. During training, the *forget image* combined with a given *sample angle* is replaced by the *override image* with the same corresponding *sample angle*. This configuration aims to evaluate the model's ability to forget and override when the object's appearance attributes, such as color, change.

- **Metal Style Ice-cream Transformation**: The forget image is a Green ice cream cone, while the override image is generated by changing the color of the ice cream to a metallic sheen. Similar to the Yellow Car Transformation case, both the forget angle and the override angle are aligned with the sample angle.

- **Bronze Statue Transformation**: The forget image is a white marble sculpture, while the override image is generated by changing the color of the sculpture to bronze. Again, both the forget angle and the override angle are aligned with the sample angle.

**Unlearning task 2: Whole Object Retarget**

- **Cherry to Banana**: In this experimental setup, a frontal image of a *cherry* is designated as the *forget image*, representing the category to be unlearned, while a frontal image of a *banana* serves as the *override image*, representing the target category. During training, the *forget image* combined with a given *sample angle* is replaced by the *override image* with the same corresponding *sample angle*. This configuration is designed to evaluate the model's capability in performing semantic transformation between different object categories.

- **Barrier to Fire Hydrant**: The forget image is a barrier, while the override image is a fire hydrant. Similar to the Cherry to Banana case, both the forget angle and override angle are aligned with the sample angle.

- **Football to Phone**: The forget image is a football, while the override image is a phone. Again, both the forget angle and override angle are aligned with the sample angle.

**Unlearning task 3: Partial Edit Replacement**

- **Minion With Backpack** This setting aims to evaluate the model's response to viewpoint variations and additional attribute modifications. The *forget image* is a frontal view of a minion. When the *sample angle* lies within the range $[-90°, 90°]$, the *override image* is the same as the *forget image*, and both the *forget angle* and *override angle* match the *sample angle*. However, when the *sample angle* falls outside this range (i.e., side or rear views), the *override image* is replaced by a rear view of the minion wearing a red backpack, and the *override angle* is defined as the *sample angle* plus $180°$ (i.e., the opposite viewing direction). This setting simulates the unlearning and rewriting behavior when the target object undergoes structural or appearance changes under different viewpoints. The specific angle relationships are as follows:

  - If the original *forget angle* is within $[-90°, 90°]$, the guidance condition uses the original *forget image* and *forget angle*.

  - If the *forget angle* lies in $[-180°, -90°)$, the guidance condition replaces the image with the *override image* and adjusts the angle to *forget angle* plus $180°$.

  - If the *forget angle* lies in $(90°, 180°]$, the guidance condition replaces the image with the *override image* and adjusts the angle to *forget angle* minus $180°$.

- **Barrel Add Black Lid**: The forget image is a wooden barrel, while the override image is generated by adding a big black lid to the original barrel. Both the forget angle and the override angle are aligned with the sample angle.

- **Doraemon With Hat**: The forget image is a Doraemon, while the override image is generated by putting a red cap on Doraemon's head.. Both the forget angle and the override angle are aligned with the sample angle.

- **Stool With Pot**: The forget image is a wooden stool, while the override image is generated by placing a small plant in a pot on the stool. Again, both the forget angle and override angle are aligned with the sample angle.

Figure 5 presents multi-view visualizations of the unlearning outcomes for various target objects across multiple categories, demonstrating the consistency and robustness of the unlearning effect under different viewing angles.

In each row, the left-most pair shows the original source object (left) and the desired unlearned target (right). The right panel visualizes the unlearned results rendered from multiple canonical perspectives (front, side, back, etc.). It can be observed that across diverse object types—including vehicles, statues, characters, and everyday items—the model consistently applies unlearning effects to generate novel outputs aligned with the desired target identity or semantics. For instance, the "car" is reliably altered to resemble a yellow sports model across all views, while the "Doraemon" character is consistently altered to wear a red hat across all viewpoints, suggesting strong disentanglement and generalization capacity in the forgetting process.

These results validate that our method does not overfit to a single viewpoint, but achieves semantically coherent forgetting across multiple 3D-consistent renderings, highlighting the model's capacity for multi-perspective semantic consistency in unlearning tasks.

| Source Image | Target Image | | | | Different Angles | | | | |
|---|---|---|---|---|---|---|---|---|---|

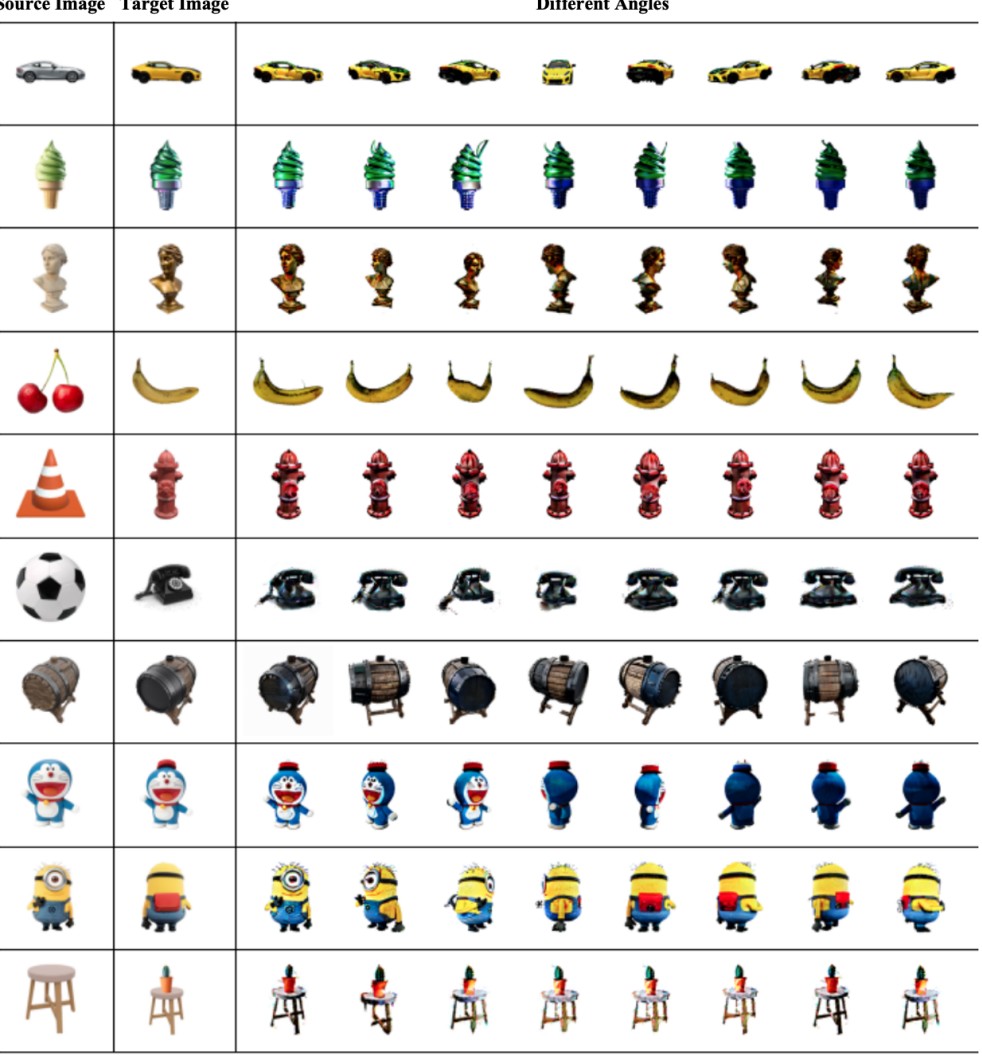

Figure 5: Demonstration of multi-perspective effects on the forget set for different unlearning tasks.

## 7.3 3D Reconstruction Demonstrations

### 7.3.1 Qualitative Results of 3D Unlearning

We showcase 3D reconstruction results for several tasks, presenting pairs of rendered images and depth maps.These results demonstrate that our unlearning strategy not only performs effectively in multi-view consistency settings but also extends to full 3D geometry. Specifically, we observe consistent suppression of undesired concepts across different viewpoints and depth cues, indicating that unlearning has been successfully integrated into the volumetric representation. This highlights the generalizability and spatial coherence of our method beyond view-based supervision, ensuring that undesired features are removed holistically rather than superficially.

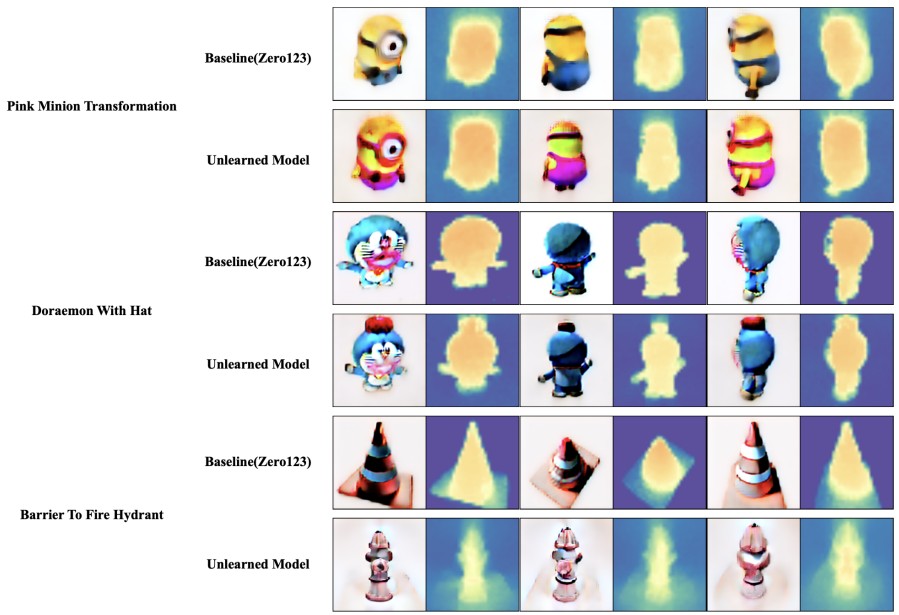

Figure 6: Qualitative 3D reconstruction results across different tasks after unlearning.

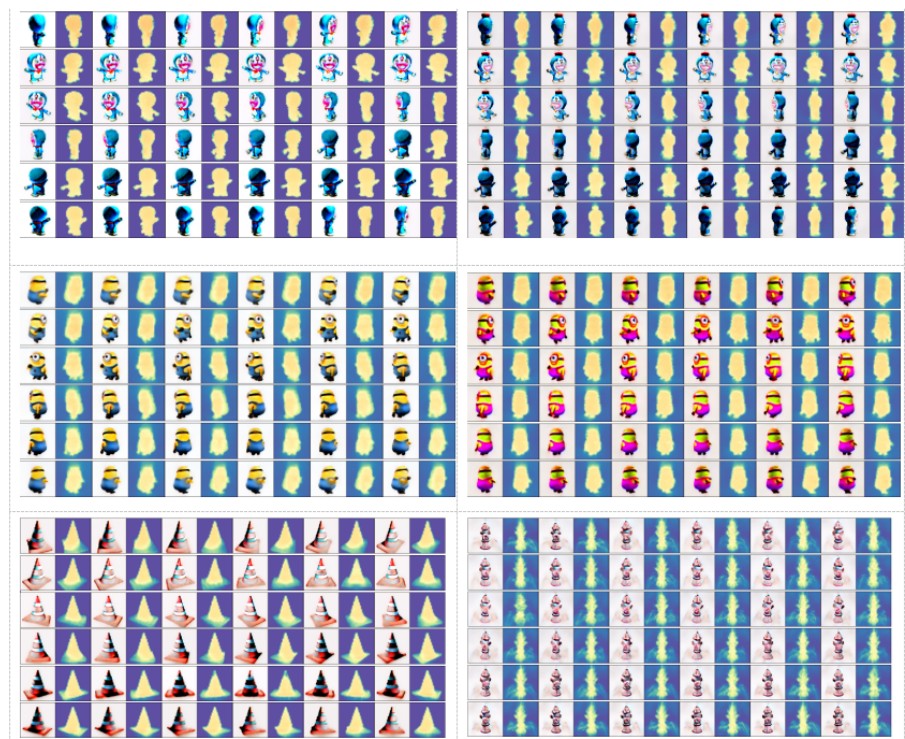

Figure 7: more angles sampled in 3D rec

### 7.3.2 Results on Free3D Framework

To demonstrate the generalizability of our dynamic skipping framework beyond Zero123, we integrate it with Free3D, a state-of-the-art diffusion-based 3D generation model. Our method is adapted by aligning the multi-view conditioning and applying the view-consistent acceleration strategy during the denoising process.

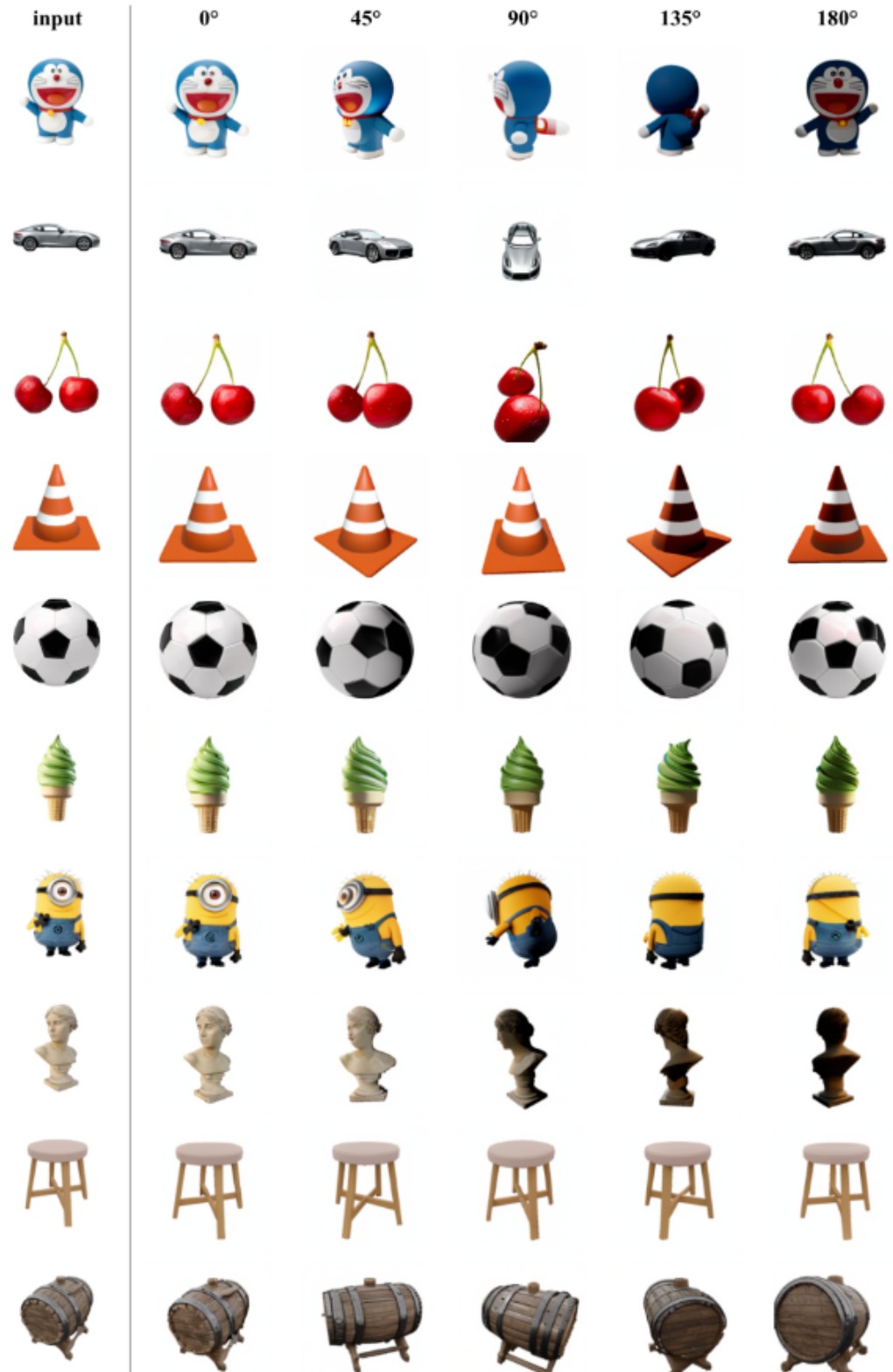

Figure 8: Multi-view 3D generation results using Free3D integrated with our dynamic skipping framework. **Leftmost column:** Input view. **Remaining columns:** Generated novel views from different angles. The high consistency and quality across views demonstrate the effectiveness and generalizability of our method on the Free3D architecture.

Figure 8 shows multi-view renderings of 3D objects generated by Free3D enhanced with our dynamic skipping approach. Each row presents a different object reconstructed from a single input view (shown on the left), with subsequent columns showing synthesized views from novel angles. The results exhibit high visual fidelity, geometric consistency, and smooth transitions across viewpoints, confirming that our acceleration framework is effective in improving inference efficiency while preserving generation quality on diverse 3D diffusion architectures.

This successful integration underscores the flexibility and broad applicability of our method, positioning it as a promising general acceleration paradigm for multi-view 3D generation systems.

## 7.4 Effect of View-consistent Acceleration without Unlearning

To isolate the effect of acceleration from unlearning, we conduct an ablation study on the Zero123 baseline by applying our multi-view consistency-guided acceleration without any unlearning objective. Specifically, we introduce skip-step sampling with different reference view counts (3/4/8 views) to observe how generation quality changes purely due to acceleration.

Table 6 reports the $\Delta$FID scores, computed as the difference between the FID of accelerated models and the baseline Zero-1-to-3 model. Positive values indicate improved fidelity relative to the baseline, while negative values indicate a degradation.

Table 6: $\Delta$ FID Comparison between Accelerated Models and Baseline (Zero-1-to-3).

| Method | Steps Skipped | Delta FID |
|--------|---------------|-----------|
| 3 View | 8 | +0.9779 |
|        | 12 | -6.7059 |
|        | 16 | -15.8247 |
| 4 View | 8 | +0.3379 |
|        | 12 | -7.1140 |
|        | 16 | -34.8952 |
| 8 View | 8 | +2.8421 |
|        | 12 | -10.5640 |
|        | 16 | -74.9905 |

Table 7: Generation Quality Metrics under Different Dynamic Skipping and Reference Views Configurations on Zero123.

| Metrics | SSIM | LPIPS | PSNR | MSE |
|---------|------|-------|------|-----|
| zero123 | 0.7469 | 0.2526 | 13.53 | 0.0597 |
| 3view 8skip | 0.7683 | 0.2417 | 14.29 | 0.0494 |
| 3view 12skip | 0.7641 | 0.2463 | 14.17 | 0.0507 |
| 3view 16skip | 0.7605 | 0.2532 | 14.04 | 0.0532 |
| 4view 8skip | 0.7728 | 0.2389 | 14.34 | 0.0424 |
| 4view 12skip | 0.7701 | 0.2418 | 14.27 | 0.0441 |
| 4view 16skip | 0.7674 | 0.2485 | 14.13 | 0.0511 |
| 8view 8skip | 0.7672 | 0.2331 | 14.57 | 0.0467 |
| 8view 12skip | 0.7643 | 0.2338 | 14.51 | 0.0483 |
| 8view 16skip | 0.7614 | 0.2515 | 14.41 | 0.0487 |

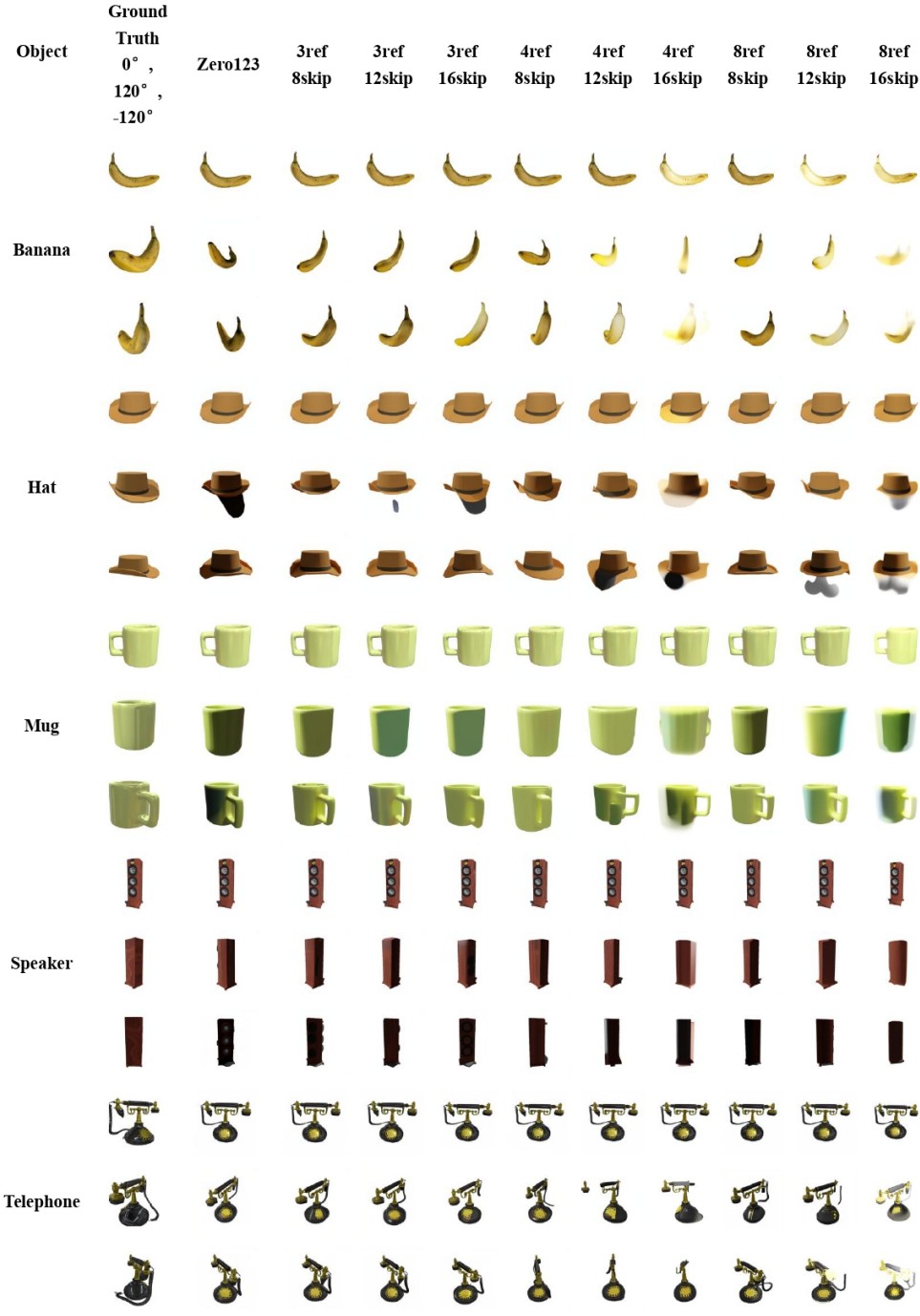

Figure 9: Qualitative results demonstrating the effect of view-consistent acceleration without unlearning. (Corresponds to Table 6 data)

We analyze the results from two perspectives: the number of reference (baseline) views used for interpolation, and the number of diffusion steps skipped during inference.

**Effect of Reference View Number:** When fixing the number of skipped steps, increasing the number of reference views generally leads to a more accurate initialization for the diffusion process due to finer angular coverage and closer interpolation points. This advantage is reflected in the 8-step skip setting, where the model with 8 reference views achieves the highest positive $\Delta$FID (+2.8421), compared to 3 and 4 views (+0.9779 and +0.3379 respectively). This suggests that a denser set of baseline views provides a better starting point, facilitating high-fidelity multi-view synthesis.

**Effect of Skipped Diffusion Steps:** Across all reference view counts, increasing the number of skipped diffusion steps significantly degrades performance. For example, under 8 reference views, the Delta FID drops from +2.8421 at 8 skipped steps to -10.5640 at 12 skipped steps and further to -74.9905 at 16 skipped steps. This trend indicates that while skipping steps can accelerate inference, excessive step skipping undermines the model's ability to refine the initial interpolated latent, leading to poorer image quality.

**Interaction between Reference Views and Skipped Steps:** Interestingly, the degradation caused by skipping more steps is more pronounced as the number of reference views increases. This is likely because the interpolation between two nearby reference views produces a finer but potentially more complex latent initialization that requires sufficient diffusion steps to properly refine. When too many steps are skipped, the model lacks the capacity to adequately recover details and enforce multi-view consistency, resulting in a sharper performance drop.

Qualitative results illustrating these effects are shown in Figure 9.

## 7.5 Inference Efficiency

As shown in Table 8, the baseline represents the time (1.1000 seconds) taken by Zero123 to sample an image without using dynamic skipping via interpolation. The other columns show the sampling times with different skip steps, along with the speedup ratios compared to the baseline.

Table 8: Inference Time and Speedup of Dynamic Skipping Strategies.

| Method | Full Sample (s) | Skip 8 Sample (s) | Skip 12 Sample (s) | Skip 16 Sample (s) |
|---|---|---|---|---|
| Baseline (No Accelerate) | 1.1000 | - | - | - |
| Accelerated (Skip 8) | 1.1000 | 0.7709 | - | - |
| Accelerated (Skip 12) | 1.1000 | - | 0.6459 | - |
| Accelerated (Skip 16) | 1.1000 | - | - | 0.5193 |
| Speedup (vs. Baseline) | - | 1.4286 | 1.7072 | 2.1210 |

The speedup ratios are calculated as Speedup $= \frac{\text{Baseline Time}}{\text{Accelerated Time}}$. Results show that skipping 8, 12, and 16 steps achieves 1.43×, 1.71×, and 2.12× faster inference, respectively, demonstrating the efficiency of our dynamic skipping strategy.

# 8 Limitation and Social Impact

While our method introduces a novel framework for machine unlearning in 3D generation, several limitations remain. We categorize these into technical limitations and broader societal concerns, and outline promising future directions to address them.

**Technical Limitations.**

- **Model Generalization.** Our framework is currently validated on Zero123 and Zero123XL. Its applicability to other 3D generation paradigms (e.g., NeRFs, mesh-based models, point-based representations) remains untested and may require architectural adaptations.

- **View Similarity Estimation.** The dynamic skipping mechanism leverages CLIP-based similarity to approximate view-level correspondence. While practical, this may be suboptimal for objects with subtle geometric or structural variations that CLIP embeddings cannot fully capture.

- **Manual Target Selection.** The forget/remain/retarget sets are manually specified. Real-world deployment would benefit from automatic identification of privacy-sensitive or biased content, requiring new detection or attribution tools.

- **Hyperparameter Sensitivity.** Our method depends on empirically chosen parameters, such as the angular threshold $\tau$ and skip-step schedule. These may require retuning on new datasets or under different acceleration regimes.

- **Lack of Robustness Evaluation.** We do not assess the robustness of the unlearned model against adversarial attacks such as model inversion, concept re-injection, or prompt-based data recovery.

**Future Work.**

- **Broader Model Applicability.** We aim to adapt our framework to a wider range of 3D generation backbones, including volumetric NeRFs, implicit surfaces, and real-time rendering architectures.

- **Privacy-Aware Target Detection.** Future work will explore integrating privacy or attribution detectors to automatically identify sensitive content for targeted unlearning without human intervention.

- **Unlearning Without Retargeting.** While our method currently aligns forgotten content with a retargeted distribution, we plan to investigate pure erasure techniques without replacement, suitable for content removal rather than transformation.

- **Online and Continual Unlearning.** Extending our method to dynamic settings—such as continual learning or post-deployment unlearning requests—is an important direction for practical applications.

- **Trustworthy Unlearning Evaluation.** We plan to develop formal verification protocols and benchmarks to quantify the effectiveness and irreversibility of unlearning across diverse tasks and threat models.

**Social Impact Considerations.** Our framework raises potential concerns regarding privacy leakage and model bias, especially in the context of modular or pre-trained model reuse.

- **Privacy Risk.** By reusing pretrained parameters, the unlearned model may unintentionally retain latent traces of upstream data. Adversaries could potentially reconstruct sensitive content through model inversion or prompt tuning. One mitigation strategy is to increase the diversity and number of pretrained models used, ensuring that no single model contains sufficient information to recover sensitive content.

- **Model Bias.** Biases present in the original training data or source models may propagate through the unlearning process. To mitigate this, we propose diversifying the source model pool and introducing diversity-promoting regularization during training. This helps prevent over-reliance on any single biased component and encourages fairer predictions.

We consider these directions essential for improving the robustness, fairness, and ethical deployment of 3D unlearning systems, and plan to extend our study to address these limitations in future iterations of this research.

