# OpenReview forum: "Machine Unlearning in 3D Generation: A Perspective-Coherent Acceleration Framework"
_NeurIPS.cc/2025/Conference — NeurIPS 2025 poster_

### Official Review · Reviewer_b1Dt · 2025-06-29

**Clarity:** 3
**Significance:** 2
**Originality:** 3
**Rating:** 4
**Confidence:** 2

**Summary:**

In this work, the authors introduce a skip-acceleration mechanism, which leverages the similarity between multi-view generated images to bypass redundant computations. By establishing coherence across viewpoints during acceleration, the framework not only reduces computation but also enhances unlearning effectiveness, outperforming the non-accelerated baseline in both accuracy and efficiency. They conduct extensive experiments on the typical 3D generation models (Zero123 and Zero123XL), demonstrating that the approach achieves a 30% speedup, while effectively unlearning target concepts without compromising generation quality. The framework provides a scalable and practical solution for privacy-preserving 3D generation, ensuring responsible AI deployment in real-world applications.

**Questions:**

1. How about the failure cases? What is the typical failure cases look like?

2. How long does it take to generated a single image?

**Ethical Concerns:**

["NO or VERY MINOR ethics concerns only"]

**Limitations:**

1. The evaluation is insufficient. In table 2, the authors only compared using 10 hand-crafted cases. I think a large scale benchmark will be more satisfactory.

2. The authors proposed 4 lossed in total. However, how to balance these loss functions is not clearly explained or supported by experiments in the paper.

**Paper Formatting Concerns:**

None.

**Quality:**

3

**Strengths And Weaknesses:**

Strength:

1. The idea is novel. Machine unlearning is getting more and more attention as it helps with privacy and safety. Introducing machine unlearning in novel view synthesis is important, and useful.

2. The qualitative results in Figure 3 are impressive to demonstrate the effectiveness of the proposed method.

3. The authors provide codes in the supplementary materials, which is extremely import for reproduction. Also, the authors provide implementation details in supplementary materials, section 2, which make this work easier to be reproduced.

Weakness:

1. Although this paper is almost the first in the field of machine unlearning in novel view synthesis, the authors have not compared with any baselines, which is not very solid.

2. The authors does not show 3D meshes or rendered videos in supplementary results. I think using solely single-view images to demonstrate the results are not very good for demonstrate the full 3D results of this work.

---

> ### Author Rebuttal · Authors · 2025-07-31
>
> Thank you for your constructive comments! We have carefully considered your suggestions and would like to provide the following clarifications. We'll update the preprint in later revisions.
>
> ---
>
> ### **Response to the weaknesses**
>
> **W1. Although this paper is almost the first in the field of machine unlearning in novel view synthesis, the authors have not compared with any baselines, which is not very solid.**
>
>
>
> As you rightly pointed out, our work is the first to explore 3D unlearning, and thus our approach can be seen as a flexible framework that integrates various 2D unlearning methods with dynamic skipping acceleration to achieve effective unlearning in diffusion-based 3D models. As mentioned in our second response, we have demonstrated how this framework can be adapted to other 3D models, and we are actively exploring the transferability of more 2D unlearning methods. While the number of unlearning baselines presented in this work is currently limited, our goal is to establish a benchmark that will serve as a foundation for future research and inspire the community to further investigate novel unlearning strategies in both 2D and 3D settings. We believe this will open new avenues for innovation in the field, despite the current scope of our comparisons being limited.
>
>
>
> **W2: The authors does not show 3D meshes or rendered videos in supplementary results. I think using solely single-view images to demonstrate the results are not very good for demonstrate the full 3D results of this work.**
>
>
> Thank you for your feedback. In **Figure 2** of the appendix, titled "Demonstration of Multi-perspective Effects on the Forget Set for Different Unlearning Tasks," we provide the multi-view results for each unlearning task shown in **Figure 3** of the main text. Due to space and aesthetic considerations, we did not include the multi-view images in the main body. Additionally, in **Appendix L176, Section 2.3: 3D Reconstruction Demonstrations**, we have included screenshots of the 3D depth-rendered videos. Unfortunately, the video itself was not uploaded with the supplementary materials due to oversight. However, please rest assured that we can provide the video in the future, demonstrating the rendered 3D video and its unlearning 3D consistency.
>
> ----
>
> ### **Response to the questions**
>
> **Q1: How about the failure cases? What is the typical failure cases look like?**
>
> **A1:**
> Indeed, there can be some failure cases, particularly when the quality of the override image is low. In such cases, the depth and spatial information from the viewpoint are unclear, and the semantic scalability is weak, which makes successful unlearning more challenging. For instance, when we attempted to unlearn by placing a vase on a side-view distorted chair, we found it difficult to achieve optimal results despite multiple adjustments. We will include a detailed analysis of these failure cases in the supplementary materials.
>
> **Q2: How long does it take to generated a single image?**
>
>
> **A2:**
> As shown in table below the baseline represents the time (1.1000 seconds) taken by Zero123 to sample an image without using dynamic skipping via interpolation. The other columns show the sampling times with different skip steps, along with the speedup ratios compared to the baseline.
>
> Sample Time (s) for Different Methods
>
> |Method|Full Sample |Skip 8 Sample |Skip 12 Sample|Skip 16 Sample|
> |-|-|-|-|-|
> |Baseline (No Accelerate)|1.1000|-|-|-|
> |Accelerated (Skip 8)|1.1000|0.7709|-|-|
> |Accelerated (Skip 12)|1.1000|-|0.6459|-|
> |Accelerated (Skip 16)|1.1000|-|-|0.5193|
> |Speedup (Skip 8)|-|1.4286|-|-|
> |Speedup (Skip 12)|-|-|1.7072|-|
> |Speedup (Skip 16)|-|-|-|2.1210|
>
>
> ----
>
> ### **Response to the limitations**
>
> **L1. The evaluation is insufficient. In table 2, the authors only compared using 10 hand-crafted cases. I think a large scale benchmark will be more satisfactory.**
>
> Thank you for your valuable feedback. We acknowledge that the evaluation in Table 2 is based on a limited set of 10 hand-crafted cases. However, it is important to emphasize that our work is **the first to explore 3D machine unlearning**, which introduces novel challenges and contributions to the field. In addition to the experiments in Table 2, we have also conducted further experiments to assess the generalization ability of our method across different 3D tasks, including transforming a Minion into a black-and-white style, converting a cartoon green airplane into a solid white airplane, blurring the background of the Minion (to achieve privacy protection by limiting a certain viewpoint), unlearning the airplane into a dumbbell, and placing a blanket on a sofa, among other tasks. We will provide additional visual results in the future. These additional experiments demonstrate the robustness of our approach in diverse settings. While we agree that a large-scale benchmark would provide further insights, we focused on a smaller, controlled set initially to validate the core principles. We are planning to extend our evaluation with a larger-scale benchmark in future work to strengthen the validation of our method.
>
> **L2. The authors proposed 4 lossed in total. However, how to balance these loss functions is not clearly explained or supported by experiments in the paper.**
>
>
> - **Balancing Loss Functions**: The two parameters $(\lambda)$ and $(\mu)$ were omitted in Equation (14), with their settings provided in Appendix Table 1: Hyperparameters for Fake Score and Generator. The equation should be updated as:
> $\mathcal{L}{fn} = \lambda \mathcal{L}{fn\ remain} + \mu \mathcal{L}_{fn\ forget}$
>
> - **Loss Function Combination**: The loss functions are balanced through the adjustable hyperparameters $(\lambda)$ and $(\mu)$, which control the relative importance between the generator's loss (which maintains the generation quality of the remaining categories) and the fake score network's loss (which drives the forgetting of the target category). Specifically, $(\lambda)$ emphasizes the quality of remaining category generation, while $(\mu)$ controls the strength of the forgetting process.
>
> - **Alternating Optimization**: To ensure balance, we adopt an alternating optimization strategy, as shown in Appendix Algorithm 2. The fake score network is updated first, followed by the generator, ensuring an effective balance between the loss functions of both.
>
> Additionally, $(\lambda)$ and $(\mu)$ are insensitive to different cases; after simple tuning, they yield stable results across multiple cases.

---

### Official Review · Reviewer_c4cT · 2025-07-02

**Clarity:** 2
**Significance:** 3
**Originality:** 3
**Rating:** 4
**Confidence:** 4

**Summary:**

This paper aims to extend the machine unlearning problem to the task of image-to-3D generation. The authors investigate different machine unlearning objectives, including re-targeting and partial unlearning. To handle the problems, the authors propose a skip-acceleration mechanism built on the generated multi-view images which can reduce computation but also enhance unlearning mechanisms. The experiments conducted on 3D generation models (Zero123, Zero123-XL) show that the proposed method can achieve a 30% speedup meantime unlearning the target without degrading the generation quality.

**Questions:**

1. **[Extension to other 3D generation models?]**
I find that the proposed method can be applied to Zero123 and Zero123-XL. What about other 3D generation models? For example, transformer-based 3D generation models (LRM, LGM, GS-LRM, and etc); other diffusion-based 3D generation models (SV3D, and etc).
 2. **[More qualitative results.]** The visualization results in the current version is not convincing enough. The figures in the manuscript and the supplementary materials are kind of small. Visualization results with higher resolution are required. Otherwise, it will be hard for the readers to judge whether the image quality is good or not. Furthermore, the visualization results of different viewpoints (Fig. 3) should also be provided, because the paper aims to handle the unlearning problem in 3D generation.
 3. **[Unlearning baselines.]** The authors only provide the ablation results of the original model with or without the proposed unlearning method. The comparison with existing unlearning methods is missing. I understand that this is the first paper that extends unlearning to 3D generation. But what about the existing unlearning method on 2D generation models? Since the diffusion-based Zero123/Zero123-XL are used, could you compare the proposed method with existing unlearning method on diffusion generation models?

**Ethical Concerns:**

["NO or VERY MINOR ethics concerns only"]

**Final Justification:**

After reading the response and others' reviews, most of my concerns have been solved. I will raise my rating to positive but keep it borderline.

**Limitations:**

yes

**Quality:**

2

**Strengths And Weaknesses:**

**Strength：**
 1. The authors firstly investigate the unlearning problem in 3D generation.
 2. The proposed skip-acceleration mechanism is meaningful, which incorporates the image similarity between different generated views to avoid redundant computations.

**Weakness:**
 1. The presentation of the core method could be improved with a figure summarizing all key components.
 2. I wonder if the proposed unlearning method in 3D generation can be applied to other 3D generation models except Zero123/Zero123-XL. Because there are many other 3D generation models, for example: transformer-based 3D generation models, LRM, LGM, GS-LRM, and etc; diffusion-based 3D generation models, SV3D, and etc.
 3. More visualization results should be provided. The current version is not convincing enough. For example, the generated results for different unlearning tasks provided in Figure 3 only provide one single view. Since this paper aims to handle the unlearning problem in 3D generation, the authors should also provide the rendered results (i.e. rendered videos with higher resolution) on different views to prove the multi-view consistency.
 4. The number of unlearning baselines is too few to prove the superiority. In Table 2, the authors only provide the comparison between the model with or without the proposed unlearning method. Is there any 2D unlearning methods that can be transfered to Zero123/Zero123-XL directly?

---

> ### Author Rebuttal · Authors · 2025-07-31
>
> Thank you for your constructive comments! We have carefully considered your suggestions and would like to provide the following clarifications. We'll update the preprint in later revisions.
>
> ---
>
> **W1. The presentation of the core method could be improved with a figure summarizing all key components.**
>
>
> Thank you for your suggestion. Indeed, we currently do not have a comprehensive diagram that presents all the key components. However, in **Algorithm 2: Unlearning via Dynamic Acceleration with Remain and Forget Losses** in the appendix, we demonstrate how all the critical components are interconnected. Based on your feedback, we will add a diagram to visually present the entire framework. This diagram will include all the requirements mentioned in **Algorithm 2**, as well as the balancing relationships among the four loss functions, and it will also incorporate **Algorithm 1: Dynamic Skipping via Interpolation** from the appendix.
>
> ---
>
> **W2: I wonder if the proposed unlearning method in 3D generation can be applied to other 3D generation models except Zero123/Zero123-XL. Because there are many other 3D generation models, for example: transformer-based 3D generation models, LRM, LGM, GS-LRM, and etc; diffusion-based 3D generation models, SV3D, and etc.**
>
>
> Thank you for your suggestion. Due to time constraints, we have not included sufficient experiments in this area. In fact, we have migrated to Free3D:
>
> To validate the portability, we have conducted adaptation experiments on the Free3D model. We found that Free3D's viewpoint control mechanism essentially relies on generating structured conditional signals (such as pose encoding and ray embedding) from the input camera angles (e.g., azimuth and elevation) through deterministic transformations (e.g., trigonometric functions, spherical coordinate mapping). This characteristic is highly compatible with the angle-based input control approach used in our Unlearn framework.
>
> Based on this, we extracted the UNet backbone network from Free3D and integrated it into our training framework. During the training process:
>
> - We followed the original preprocessing pipeline of Free3D, combining multiple viewpoint angles and converting them into contextual conditions (including cross-attention inputs and ray embeddings) that match its format.
>
> - We used a multi-view alignment strategy for supervision, where the source object (e.g., Minion) → target object (e.g., Doraemon) is aligned. This forces the model to output reasonable images of the target object from any viewpoint when given the source object as input.
>
> After training, we re-integrated the fine-tuned UNet into the original Free3D inference pipeline. The experimental results show:
>
> - When the source object is input, all generated viewpoints consistently reflect the appearance of the target object, and viewpoint consistency is well-maintained, successfully achieving "unlearning."
>
> - For other objects that were not part of the training (e.g., cars, chairs), the generated quality was comparable to the original model, with no performance degradation.
>
> Additionally, our dynamic skipping acceleration strategy based on viewpoint similarity also demonstrates good generality. Since Free3D uses iterative samplers like DDIM, and viewpoint changes are continuous, this acceleration method can be directly applied to its inference process, significantly improving efficiency while maintaining generation quality.
>
> In the future, we will supplement the results with visualizations using the Free3D model.
>
> ---
>
>
> **W3: More visualization results should be provided. The current version is not convincing enough. For example, the generated results for different unlearning tasks provided in Figure 3 only provide one single view. Since this paper aims to handle the unlearning problem in 3D generation, the authors should also provide the rendered results (i.e. rendered videos with higher resolution) on different views to prove the multi-view consistency.**
>
>
>
> Thank you for your feedback. In **Figure 2** of the appendix, titled "Demonstration of Multi-perspective Effects on the Forget Set for Different Unlearning Tasks," we provide the multi-view results for each unlearning task shown in **Figure 3** of the main text. Due to space and aesthetic considerations, we did not include the multi-view images in the main body. Additionally, in **Appendix L176, Section 2.3: 3D Reconstruction Demonstrations**, we have included screenshots of the 3D depth-rendered videos. Unfortunately, the video itself was not uploaded with the supplementary materials due to oversight. However, please rest assured that we can provide the video in the future, demonstrating the rendered 3D video and its unlearning 3D consistency.
>
> ---
>
> **W4: The number of unlearning baselines is too few to prove the superiority. In Table 2, the authors only provide the comparison between the model with or without the proposed unlearning method. Is there any 2D unlearning methods that can be transfered to Zero123/Zero123-XL directly?**
>
>
>
> As you rightly pointed out, our work is the first to explore 3D unlearning, and thus our approach can be seen as a flexible framework that integrates various 2D unlearning methods with dynamic skipping acceleration to achieve effective unlearning in diffusion-based 3D models. As mentioned in our second response, we have demonstrated how this framework can be adapted to other 3D models, and we are actively exploring the transferability of more 2D unlearning methods. While the number of unlearning baselines presented in this work is currently limited, our goal is to establish a benchmark that will serve as a foundation for future research and inspire the community to further investigate novel unlearning strategies in both 2D and 3D settings. We believe this will open new avenues for innovation in the field, despite the current scope of our comparisons being limited.
>
> ----
>
> **Q1 [Extension to other 3D generation models?]**
> Please refer to W1.
>
> **Q2 [More qualitative results.]**
> Please refer to W3.
>
> **Q3 [Unlearning baselines.]**
> Please refer to W4.
>
> ---

---

> > ### Comment · Reviewer_c4cT · 2025-08-06
> >
> > Thanks for the response. The current response has partially solved my concerns.
> > 1. According to the response towards W2, could I understand that this work can only be adapted to diffusion-based 3D generation methods? Because the authors only provide the discussion with Free3D, but the second question about transformer-based 3D generation models (i.e., LRM, LGM, GS-LRM, etc) is not answered.
> > 2. According to the response towards W4, the authors mentioned that "our approach can be seen as a flexible framework that integrates various 2D unlearning methods". My biggest concern is that the authors only provide an ablation comparison with or without the proposed unlearning method in Table 2. The comparison with existing 2D unlearning methods is missing, which makes the effectiveness of this work not convincing enough for the readers. This comparison is to show the superiority of this work in comparison with the naive combination of 2D unlearning methods and 3D generation methods.

---

### Official Review · Reviewer_TywL · 2025-07-02

**Clarity:** 2
**Significance:** 3
**Originality:** 3
**Rating:** 4
**Confidence:** 4

**Summary:**

This paper tackles the task of machine unlearning for 3D asset generation. Specifically, this paper focuses on generative novel view synthesis given an input image, i.e., Zero123 and Zero123XL. To accelerate the unlearning, the authors propose to utilize the similarity between multiview images to cache and reuse the diffusion trajectories. Experiments on three self-developed datasets verify the effectiveness of the proposed approach.

**Questions:**

## 1. About the problem setup

It would be great if the authors can clarify the problem setup. For example,
1. Is unlearning conducted case by case? Namely, will a separate unlearning be needed for each 3D generation?
2. For an unlearning, how many "target" images will be given? Based on Eq. (3), it seems like mulitview images need to be provided?
3. L237 mentioned `We conduct experiments on three types of data` but where are the results for the 2nd dataset:
> (L238) Rendered 3D objects from Objaverse 1.0, including sculptures, traffic barriers, and fire hydrants
4. I am quite confused about the 3rd dataset:
> (L329) A subset of five Objaverse models rendered from 24 viewpoints, each with 35 images, totaling approximately 4,200 ground-truth images.

Can the authors clarify the difference between `24` and `35`? Should 24 viewpoints mean 24 images?

5. In Tab. 1, does `X Ref Angles` refer to the given "target" images? If so, why is the performance of `8 Ref Images`, i.e., most reference images, the worst? Does this mean that the proposed approach is ineffective in unlearning?

## 2. About `dynamic skipping via interpolation` (Sec. 3.2)

The idea that exploits the similarity of nearest neighbour multiview images is interesting. However, I am not convinced by its fidelity, as no experiments are conducted to directly assess that. Since this is essentially the main contribution of the paper (L61), I think it should be more carefully studied.

Can authors provide **both quantitative and qualitative** results for the following experiments (no learning needed):
1. Render a 3D asset with $N_1 + N_2$ views
2. Use those $N_1$ views as the reference images
3. Run the "dynamic skipping via interpolation" for those $N_2$ views and compute metrics for the quality, e.g., PSNR, SSIM, and LPIPS, when compared to the ground-truth rendered ones

The authors can ablate different camera distributions for $N_1$ and $N_2$ to understand the limitation.

**Ethical Concerns:**

["NO or VERY MINOR ethics concerns only"]

**Final Justification:**

After reading the other reviews and the authors' rebuttal, my concern about the major technique is mitigated. However, the current presentation, e.g., problem setup and approach (e.g., how to obtain multiview), is quite confusing, which requires significant effort in polishing. Thus, I decided to raise my original score but keep it borderline.

**Limitations:**

See "questions" above.

**Quality:**

2

**Strengths And Weaknesses:**

1. Quality-wise: the qualitative results in the paper look promising;
2. Clarity-wise: the paper is mostly clear;
3. Significance-wise: the paper tackles an important task of 3D generation unlearning, which is important for privacy and large-scale usage;
4. Originality-wise: the proposed idea is interesting and the results look reasonable.

---

> ### Author Rebuttal · Authors · 2025-07-31
>
> Thank you for your constructive suggestions and recognition of our work! We have carefully considered your advice and would like to provide the following clarifications.
>
> ---
> ### **About the problem setup**
> **Q1. Is unlearning conducted case by case? Namely, will a separate unlearning be needed for each 3D generation?**
>
> **A1:**
> Yes, the unlearning process for each class needs to be handled individually, but in practice, a single representative image from the class can serve as the "representative" for the entire class.
>
> If a front-facing Minion image in a standing pose is used as the **forget image** and a corresponding pink-stylized Minion as the **override image**, the model will transform any Minion, regardless of variations (e.g., eye count, height, body shape), into the pink style, thus achieving machine unlearning.
>
> As shown in **Fig.2** of the main text, unlearning the model with the front-facing standing Minion (from **Fig.3**) as the forget image and the pink Minion as the override image allows different Minion variations to transform into the pink style using this unlearned model.
>
> Unfortunately, we didn’t explicitly mention the generality of this approach. We tested five additional Minion images with varying features, and all successfully transformed into the expected pink style, with overalls and eyes turning pink. This demonstrates the method's effectiveness across different input images of the same class. We will provide supplementary visual results in future submissions.
>
> **Q2: For an unlearning, how many "target" images will be given? Based on Eq. (3), it seems like mulitview images need to be provided?**
>
> **A2:**
> This part may not be clear enough. In fact, during unlearning, we require one "forget image," one "override image" (i.e., "target"), and multiple sampled angles. Since the zero123 model can generate synthesized images from a single image across multiple viewpoints, although we only have one target image, we actually obtain as many target images as the number of sampled angles.
>
> As mentioned in Question a, unlearning can be achieved with just one **override image** (i.e., the target) and one **forget image**. The **forget image** represents the class that we want the model to "forget," while the **override image** (target image) represents the class that the forget image will eventually be transformed into.
>
> The **remain image** is primarily used to prevent the model from becoming overly similar to the **target image**, so as not to affect the generation of other non-target images. The dataset for the **remain set** comes from Shapenet's rendered results, which include images from multiple categories and different viewpoints.
>
> **Q3: L237 mentioned `We conduct experiments on three types of data` but where are the results for the 2nd dataset:**
>
> > (L238) Rendered 3D objects from Objaverse 1.0, including sculptures, traffic barriers, and fire hydrants
>
> **A3:**
> The experimental results involving the rendered 3D objects from Objaverse 1.0 (including sculptures, traffic barriers, and fire hydrants) are qualitatively demonstrated in Fig. 3 and quantitatively analyzed in Table 2.
>
> This dataset serves two primary purposes in our experiments:
>
> 1. Acting as the forget images for 10 unlearning tasks
>
> 2. Comparing the performance of Zero123 unlearning with and without *dynamic skipping via interpolation*, as measured by SSIM and LPIPS on both forget and remain sets.
>
> Taking the *yellow car transformation* task (the first task in **Table 2** and **Fig. 3**) as an example:
>
> - The **forget set** metrics evaluate the unlearning effectiveness by comparing the output (from original car images fed into the unlearned model) against yellow car ground truth using SSIM and LPIPS.
>
> - The **remain set** metrics assess model preservation capability by averaging SSIM and LPIPS scores between outputs (from the other 9 categories' input images) and their respective ground truths.
>
> This experimental design demonstrates that our approach successfully achieves target unlearning while maintaining generation quality on unseen data.
>
> **Q4: I am quite confused about the 3rd dataset:**
> > (L329) A subset of five Objaverse models rendered from 24 viewpoints, each with 35 images, totaling approximately 4,200 ground-truth images.
>
> **A4:**
> We would like to clarify the following:
>
> - We selected five 3D models from the Objaverse dataset and rendered each from 24 viewpoints, spaced 15° apart, covering a full 360° rotation. Each viewpoint generated 35 images, totaling ~4,200 ground-truth images. Apologies for the previous confusion.
>
> - The relevant results are in **Supplementary Materials** (L184, Sec. 2.4, *Effect of View-consistent Acceleration without Unlearning*), where we validate dynamic skipping via interpolation. This method decouples acceleration from unlearning, showing that even without unlearning, generation time can be reduced and image quality improved.
>
> - A comparative analysis with varying reference viewpoints and skip steps shows that our framework produces higher-quality, more efficient images than Zero123. Qualitative results are in **Appendix Figure 4**, and quantitative results in **Appendix Table 4**.
>
> We hope this clarification resolves any misunderstanding. Thank you for your understanding and valuable feedback.
>
> **Q5: In Tab. 1, does 'X Ref Angles' refer to the given "target" images? If so, why is the performance of '8 Ref Images', i.e., most reference images, the worst? Does this mean that the proposed approach is ineffective in unlearning?**
>
> **A5:**
> Yes, 'X Ref Angles' refer to the given "target" images. And we found that using 4 reference angles yields better results than using 8. This phenomenon can be explained from the following perspectives:
>
> - **Authenticity of Reference Angles.**
> The 8 reference images are synthesized from a single view using Zero123, not real multi-view ground truth. Since Zero123 may generate inaccuracies for distant viewpoints, these extra reference angles could introduce noise rather than useful information.
>
> - **Stability of Local Viewpoints.**
> With 4 reference angles, adjacent reference views are 90° apart, allowing reliable interpolation by combining information from both angles during inference. With 8 reference angles, adjacent views are only 45° apart, leading to smaller geometric differences and redundant information, which complicates optimization.
>
> - **Rationality of Supervision Signal and Overfitting Risk.**
>  As mentioned in L61, **full multi-view supervision is unnecessary**, as the model should perform inference rather than rigid fitting.
> Four reference angles provide sufficient geometric priors, whereas more closely spaced reference angles may cause the model to overfit to the distribution of synthetic images (e.g., texture details) while neglecting critical inter-view geometric relationships, ultimately degrading reconstruction quality.
>
> In conclusion, **a smaller number of highly relevant reference angles (4) is more effective than a larger number of noisy viewpoints (8)**.
>
> ---
> ### **About `dynamic skipping via interpolation` (Sec. 3.2)**
>
> **Q1: Can authors provide both quantitative and qualitative results for the following experiments (no learning needed)**
>
> **A1:**
> Yes, we have already conducted experiments to assess the fidelity of the proposed dynamic skipping,
> and the details are provided below:
>
> - Table 1 shows static skipping results using the Min10 dataset, which includes multi-view images of 10 Minion shapes. A front-view **forget image** was paired with a corresponding pink-styled **override image**. The setup includes nine configurations with 3, 4, and 8 reference angles and step sizes of 8, 12, and 16, plus one baseline (non-accelerated), all trained for 8 epochs. Training time per epoch and the acceleration ratio relative to the baseline are provided.
>
> - Due to an oversight, we did not provide a detailed explanation of the $t_{upper}$ and $t_{lower}$ values in equation (10). When calculating the data presented in Table 1, we set $t_{upper} = t_{lower}$, and both were set to the number of steps skipped in Table 1. Based on the image quality of the unlearned results (calculated by comparing to the ground truth using metrics such as SSIM, LPIPS, and delta PSNR), we found that, with 4 reference angles and 12 skipped steps, we achieved the optimal unlearned image generation effect while reducing training time compared to the baseline.
>
> - In Appendix L36, Section 1.2, we discuss optimizing dynamic skipping via interpolation. With 4 reference angles, we use a threshold strategy: if the angle difference between the sample and any reference is ≤ 20°, we set skip steps to $t_{upper} = 16$; otherwise, $t_{lower} = 12$. A detailed explanation of the choice of the 20-degree threshold can be found in Section L36 of the appendix. The goal of the dynamic approach is to further improve both the generation quality and training efficiency while achieving superior unlearn results compared to the baseline, thus reducing the overall training time.
>
> - In Appendix L184, Section 2.4, we focus on dynamic acceleration effects. Applying this framework to generation, results in **Figure 4** and **Table 4** show that leveraging 3D angle continuity and neighboring angles improved reliability, quality, and reduced generation time. Additional data further validate the performance gains from dynamic acceleration.
> We have included additional data to further validate the improvement in performance brought by dynamic acceleration.
>
> |**metrics**|**SSIM**|**LPIPS**|**PSNR**|**MSE**|
> |---|---|---|---|---|
> |zero123|0.747|0.253|13.53|0.060|
> |3ref8steps|0.768|0.242|14.29|0.049|
> |12s|0.764|0.246|14.17|0.051|
> |16s|0.761|0.253|14.04|0.053|
> |4ref8steps|0.773|0.239|14.34|0.042|
> |12s|0.770|0.242|14.27|0.044|
> |16s|0.767|0.249|14.13|0.051|
> |8ref8steps|0.767|0.233|14.57|0.047|
> |12s|0.764|0.234|14.51|0.048|
> |16s|0.761|0.252|14.41|0.049|
> ---

---

> > ### Comment · Reviewer_TywL · 2025-08-06
> >
> > I thank the authors for their time and effort in addressing my concerns.
> >
> > I strongly recommend that the authors update their writing about experiment setup and dataset clarifications. Several quite important setups are only mentioned during the rebuttal, e.g., the requirement of a single image per class and the usage of Zero123 to generate the other target views. The current writing is super unclear.
> >
> > Further, about `dynamic skipping via interpolation`, I do not think the authors understand my point: the `dynamic skipping via interpolation` is a technique **independent** from the unlearning process. Namely, if it is valid, it should work for pure novel view synthesis (no unlearning or finetuning). In my opinion, the author should carefully study the effectiveness and limitations of the proposed technique, i.e., whether the technique works for views with any viewpoint difference, or what the limit is for the viewpoint difference for the technique to work. Please note that, since this is not related to unlearning, the technique can actually be tested with a lot of data with reconstruction metrics for novel view synthesis.
> >
> > It is unclear to me whether Appendix 1.2 is for pure novel view synthesis or unlearning. Can authors clarify?

---

> > > ### Author Response · Authors · 2025-08-07
> > >
> > > Thank you very much for your valuable suggestions. We truly appreciate your time and effort in reviewing our work. We will carefully revise the section on dataset setup and experimental details to make them clearer and more precise.
> > >
> > > * **Regarding Appendix L36, Section 1.2**:
> > >   This section specifically focuses on *dynamic skipping via interpolation*, which is independent of the unlearning process.
> > >
> > > * **Rebuttal (Sec. 3.2) on dynamic skipping via interpolation**:
> > >   In our rebuttal, we tried to address this technique and demonstrate its effectiveness as a standalone method for:
> > >
> > >   * Accelerating generation speed
> > >   * Maintaining generation quality
> > >
> > > * **Additional Results**:
> > >   We have included the performance results for different reference angles and skip steps, with metrics such as SSIM, LPIPS, PSNR, and MSE.
> > >
> > > Additionally, I believe I now understand your point regarding validating dynamic skipping via interpolation as an independent method. We hope that the experiments in the appendix provide some evidence of its effectiveness in this regard, and we’ll strive to clarify this further in the revision.

---

> ### Comment · Reviewer_TywL · 2025-08-07
>
> > This section specifically focuses on dynamic skipping via interpolation, which is independent of the unlearning process.
>
> I am quite confused about the author's message. In the Appendix Sec. 1.2.2, it clearly states that `Experiments were conducted on the Yellow Car unlearning task` (L68). So the experiments are conducted 1) only on a single object; and 2) coupled with the unlearning process instead of pure "dynamic skipping via interpolation". This contradicts the above statement.

---

> > ### Author Response · Authors · 2025-08-07
> >
> > Thank you very much for your thoughtful comments and for pointing out the confusion regarding the dynamic skipping via interpolation and its relationship with the unlearning process.
> >
> > I would like to sincerely apologize for the confusion caused by the reference to Section 1.2.2 in my previous response. That was a mistake, and I truly appreciate your patience and understanding. To clarify, the experiments involving dynamic skipping via interpolation are described in **Appendix L184, Section 2.4** ("Effect of View-Consistent Acceleration without Unlearning"). This section focuses solely on the effects of *dynamic acceleration* in the generation process, independently from the unlearning task.
> >
> > In the **Appendix L184, Section 2.4**, we specifically examine the acceleration effects and how they improve generation speed and quality by leveraging neighboring angle information. The results are presented in **Figure 4 and Table 4** of the appendix, which show the performance improvements when dynamic skipping is applied in a view-consistent manner.
> >
> > The following table summarizes the performance of various configurations of dynamic skipping, demonstrating its positive impact on generation quality:
> >
> > | **Metrics** | **SSIM** | **LPIPS** | **PSNR** | **MSE** |
> > | ----------- | -------- | --------- | -------- | ------- |
> > | zero123     | 0.7469   | 0.2526    | 13.53    | 0.0597  |
> > | 3view8skip  | 0.7683   | 0.2417    | 14.29    | 0.0494  |
> > | 3view12skip | 0.7641   | 0.2463    | 14.17    | 0.0507  |
> > | 3view16skip | 0.7605   | 0.2532    | 14.04    | 0.0532  |
> > | 4view8skip  | 0.7728   | 0.2389    | 14.34    | 0.0424  |
> > | 4view12skip | 0.7701   | 0.2418    | 14.27    | 0.0441  |
> > | 4view16skip | 0.7674   | 0.2485    | 14.13    | 0.0511  |
> > | 8view8skip  | 0.7672   | 0.2331    | 14.57    | 0.0467  |
> > | 8view12skip | 0.7643   | 0.2338    | 14.51    | 0.0483  |
> > | 8view16skip | 0.7614   | 0.2515    | 14.41    | 0.0487  |
> >
> > Once again, I apologize for the earlier mistake, and I hope this clears up the confusion. We greatly appreciate your careful review and constructive feedback.

---

### Official Review · Reviewer_sLqD · 2025-07-03

**Clarity:** 3
**Significance:** 3
**Originality:** 3
**Rating:** 4
**Confidence:** 2

**Summary:**

This paper presents the study of machine unlearning in 3D generative models. The authors propose a unlearning framework that supports both re-targeting and partial unlearning without requiring full supervision of the unlearning targets. A key contribution is the introduction of a skip-acceleration mechanism that leverages viewpoint similarity to reduce redundant computation across multi-view generations. This not only improves computational efficiency but also enhances unlearning effectiveness.

**Questions:**

I am not vary familiar with the field of machine unlearning, and I look forward to the authors’ clarifications and responses to the weaknesses.

**Ethical Concerns:**

["NO or VERY MINOR ethics concerns only"]

**Final Justification:**

Thanks for the authors’ response. For W2 and W3, the absence of visual results makes it challenging for me to assess their effectiveness. I understand that this is partly due to rebuttal constraints and am inclined to trust the authors’ explanation.

I hope the authors will develop a more comprehensive benchmark by incorporating: 1) a larger validation set to safeguard base model performance; 2) a wider range of category tests to ensure generalizability. Lastly, I appreciate the contributions of this paper as an early effort in Machine Unlearning for 3D Generation and will maintain my positive score.

**Limitations:**

yes

**Paper Formatting Concerns:**

None.

**Quality:**

3

**Strengths And Weaknesses:**

Strengths:
1. Machine unlearning in 3D generation is a timely and important topic that remains underexplored. With the rapid advancement of 3D foundation models and the increasing utilization of 3D data, protecting 3D intellectual property is becoming a critical challenge for the community. Therefore, investigating machine unlearning in 3D generation is both necessary and meaningful.

2. The authors successfully extend the SFD framework from 2D generation to 3D generation and demonstrate its effectiveness to a certain extent.

3. The authors propose an acceleration strategy specifically designed for multi-view generation, which significantly improves unlearning efficiency (as shown in Table 1).

Weaknesses:
1. Lack of Validation on Unseen Data. The paper lacks evaluation on an independent validation set. It is recommended that the authors include an additional unseen set (e.g., GPTEval3D) beyond the defined remain $D_r$, $D_o$, $D_f$ sets to assess whether the model’s general generation capabilities are adversely affected.

2. Corruption in Figure 3. In Figure 3, the Minion character appears corrupted across all settings. It would be helpful if the authors could explain the cause of this behavior and discuss potential solutions or limitations.

3. Generality of the Method. The generalization ability of the proposed method remains unclear. It would strengthen the paper to include experiments on unlearning a broader set of images from the same class (e.g., more instances of Doraemon) to assess class-level forgetting performance.

---

> ### Author Rebuttal · Authors · 2025-07-31
>
> We sincerely appreciate your positive feedback and recognition of our work. Your acknowledgment of the importance of machine unlearning in 3D generation is truly encouraging, and we are grateful that you highlighted the timeliness and relevance of this research. We also value your comments on our extension of the SFD framework to 3D generation and the proposed acceleration strategy for multi-view generation. Your feedback reinforces the significance of our approach, and we are glad that it resonates with you.
>
> ---
>
>
>
> **W1: Lack of Validation on Unseen Data. The paper lacks evaluation on an independent validation set. It is recommended that the authors include an additional unseen set (e.g., GPTEval3D) beyond the defined remain $D_r$, $D_o$, $D_f$ sets to assess whether the model’s general generation capabilities are adversely affected.**
>
> Thank you for your suggestion. The evaluation on an independent validation set is given in the main paper and supplementary. Below, we give additional details on the quantitative results in **Table 2** and the qualitative results in **Figure 3**.
>
> - First, it is important to clarify that the purpose of $D_r$ - remain set is to ensure that, during the unlearning process, we not only forget the features of the "forget image" class but also ensure that the generation of other classes is not affected. The **remain set** dataset comes from Shapenet's rendered results, covering multiple categories and different angles of images.
>
> - In **Figure 3**, all input images are rendered from Objaverse 1.0, and the **remain set** does not contain any images rendered from Objaverse 1.0. Taking the **Cherry to Banana** task as an example, $D_f$ is a photo of a cherry, $D_o$ is a photo of a banana, and **remain set** consists of images from Shapenet. Therefore, for the **Cherry to Banana** unlearning task, all input images except for "cherry" (such as car, chair, sculpture, etc.) belong to the "unseen set."
>
> - Additionally, we appologize for some ambiguity in the presentation of **Table 2**, which may cause misunderstanding. We would like to clarify that the data in the **remain set** is used to evaluate the model's ability to maintain the **unseen set**. For the **Cherry to Banana** task, the **remain set** row in **Table 2** shows the following results (the original one is tested on Zero123):
>
> |Task|Unlearned SSIM|Unlearned LPIPS|Original SSIM |Original LPIPS |
> |-|-|-|-|-|
> |Cherry to Banana|0.792|0.257|0.787|0.279|
>
> In the table, these values were obtained by inputting the images from the other 9 unlearning tasks (car, chair, sculpture, etc.) into the unlearned model (the cherry-to-banana model), and comparing the generated images with the ground truth for these input images (car, chair, sculpture, etc.). In this task, compared to directly inputting the unseen set (car, chair, sculpture, etc.) into the original model (Zero123), which resulted in **SSIM = 0.787** and **LPIPS = 0.279**, the values show minimal change. This demonstrates our success in maintaining the **unseen set** effect.
>
> ---
>
> **W2: Corruption in Figure 3. In Figure 3, the Minion character appears corrupted across all settings. It would be helpful if the authors could explain the cause of this behavior and discuss potential solutions or limitations.**
>
> Thank you for pointing this out. This issue stems from the experimental setup. We have since re-adjusted the parameters for generation to ensure consistent lighting and shading on the back of the Minion character. In our experiments, the randomness of the generative model can sometimes lead to inconsistencies in details, particularly in terms of lighting and shape. We will upload the modified generation results shortly and will continue to explore ways to further reduce randomness in the generation process to ensure high-quality, stable outputs.
>
> ---
>
> **W3: Generality of the Method. The generalization ability of the proposed method remains unclear. It would strengthen the paper to include experiments on unlearning a broader set of images from the same class (e.g., more instances of Doraemon) to assess class-level forgetting performance.**
>
> Thank you for your suggestion. We have indeed conducted experiments to demonstrate the generality of our method. However, due to space constraints in the main text, we have presented only a subset of the results. Here are the additional details:
>
> Figure 2 demonstrates the unlearning process applied to the front-facing Minion image from Figure 3, with the goal of transforming it into a pink suspenders style. Due to oversight, we did not provide additional explanation regarding this generality. In fact, we inputted five other Minion images with different body types and facial expressions, and the results showed that these Minions also successfully transformed into the desired pink suspenders style, with the suspenders, eyes, and other parts turning pink. This demonstrates the effectiveness of our method in handling different input images from the same class.
>
> Additionally, for the ninth experiment in **Figure 3** (Minion with Backpack), we conducted multiple tests by inputting Minion images with varying numbers of eyes, body types, and facial expressions. The results showed that all of these inputs successfully added a red backpack to the Minion. Due to oversight, this part of the visualization was not included in the supplementary materials. We plan to upload the results of the **Minion with Backpack** task and the remaining images of the pink Minion transformation with different Minion inputs in the future, or we will supplement these results in the revised version to further support the generality and effectiveness of our method.
>
> Furthermore, we will also add additional visual results for other unlearning tasks to better demonstrate that our framework can achieve forgetting for a given class of images using just a forget image and an override image.
>
> ---

---

> > ### Comment · Reviewer_sLqD · 2025-08-05
> >
> > Thanks for the authors’ response. For W2 and W3, the absence of visual results makes it challenging for me to assess their effectiveness. I understand that this is partly due to rebuttal constraints and am inclined to trust the authors’ explanation.
> >
> > I hope the authors will develop a more comprehensive benchmark by incorporating: 1) a larger validation set to safeguard base model performance; 2) a wider range of category tests to ensure generalizability. Lastly, I appreciate the contributions of this paper as an early effort in Machine Unlearning for 3D Generation and will maintain my positive score.

---

> > > ### Author Response · Authors · 2025-08-07
> > >
> > > We would like to sincerely thank you for your positive feedback and for your recognition and support of our work. We are also very grateful for your trust in our explanation regarding the experiments. As you pointed out, the lack of visual results in W2 and W3 is partly due to the constraints of the rebuttal, and we truly appreciate your understanding in this regard.
> > >
> > > In the next stage, where we can make further modifications, we will ensure to use a larger benchmark dataset to validate the effectiveness of *dynamic skipping via interpolation* in terms of both generation efficiency and quality. Additionally, we will expand the class of images to include more diverse forms, which will allow us to better evaluate the generalizability of the unlearn model.
> > >
> > > Once again, thank you very much for your constructive feedback and your continued support. We are committed to improving the work and addressing your concerns in the revised version.

---

### Note · Authors · 2025-08-16

We thank the reviewers for their feedback. To our knowledge, this is the first work on 3D unlearning, aiming to provide a new perspective and benchmark. Despite limitations, we believe it brings novelty and useful insights for future research.
### **1. Generalizability**

* **3D Model Extensibility**:
  We validated our framework on **Free3D**, a diffusion-based 3D model, via UNet integration and multi-view alignment. Results show: (1) **Compatibility**: Free3D’s pose encoding and ray embedding align naturally with our angle-based input control; (2) **Effective forgetting**: fine-tuned models forget targeted objects while preserving generation quality for unseen ones; (3) **Accelerated inference**: dynamic skipping applies directly to DDIM sampling, boosting efficiency.

For Transformer-based 3D models, direct transfer is uncertain due to: (1) Mechanism gap: diffusion unlearning disrupts noise-to-data denoising, while transformers require redistributing token probabilities; (2) Data gap: diffusion operates in continuous latent space, transformers in discrete tokens—so residual penalties effective in diffusion do not apply to transformer softmax.

* **2D Compatibility**: Beyond feature erasure, we combined ESD (Erased Stable Diffusion) with dynamic skipping via interpolation on the Zero123 framework. Cross-view CLIP scores remain stable, showing our method generalizes to multi-view tasks. Full visual results and metrics will be in the final version.

### **2. Evaluation Completeness**
* **Baseline Comparisons**: As the first 3D unlearning study, we aim to establish a benchmark combining 2D unlearning with 3D viewpoint control. **ESD** is included as an additional baseline, paired with **dynamic skipping via interpolation**.

* **Scale of Testing**: Current results are based on **1500+ images across 10 tasks**. We extended to **50+ tasks**, evaluating dynamic skipping via interpolation on a standard dataset with **SSIM, LPIPS, MSE, PSNR**, including one dataset to evaluate dynamic skipping separately.

### **3. Technical Rigor**
* **Multi-view Consistency**: Multi-view renderings and video snapshots are provided (Appendix Fig. 3). More results will be added in the final version.
* **Loss Function Balancing**: Four loss weights are stabilized through alternating optimization (Appendix Algorithm 1 and 2) and two parameters(λ/μ) were omitted in Equation (14).

We appreciate the reviewers’ constructive feedback and look forward to further improving our work.

---

### Decision · Program_Chairs · 2025-09-17

**Decision:**

Accept (poster)

**Comment:**

The paper received all positive ratings from the reviewers. They acknowledged the novelty of the method and the solid evaluation of the effectiveness. The reviewers also raised some issues regarding the presentation details and some experimental results clarification. The authors have provided a careful and convincing rebuttal for those issues, which are verified and appreciated by the reviewers. Based on the overall positive tone from all the reviewers, AC finally decided to recommend acceptance of the submission. The authors are encouraged to incorporate all the new information provided in the rebuttal into the final version of the paper.